# UAF1 deubiquitinase complexes facilitate NLRP3 inflammasome activation by promoting NLRP3 expression

Hui Song[1,2,4], Chunyuan Zhao[1,2,3,4], Zhongxia Yu[1,2], Qizhao Li[1,2], Rongzhen Yan[1,2], Ying Qin[1,2], Mutian Jia [1,2] & Wei Zhao [1,2 ✉]

NOD-like receptor protein 3 (NLRP3) detects microbial infections or endogenous danger signals and activates the NLRP3 inflammasome, which has important functions in host defense and contributes to the pathogenesis of inflammatory diseases, and thereby needs to be tightly controlled. Deubiquitination of NLRP3 is considered a key step in NLRP3 inflammasome activation. However, the mechanisms by which deubiquitination controls NLRP3 inflammasome activation are unclear. Here, we show that the UAF1/USP1 deubiquitinase complex selectively removes K48-linked polyubiquitination of NLRP3 and suppresses its ubiquitination-mediated degradation, enhancing cellular NLRP3 levels, which are indispensable for subsequent NLRP3 inflammasome assembly and activation. In addition, the UAF1/USP12 and UAF1/USP46 complexes promote NF-κB activation, enhance the transcription of NLRP3 and proinflammatory cytokines (including pro-IL-1β, TNF, and IL-6) by inhibiting ubiquitination-mediated degradation of p65. Consequently, *Uaf1* deficiency attenuates NLRP3 inflammasome activation and IL-1β secretion both in vitro and in vivo. Our study reveals that the UAF1 deubiquitinase complexes enhance NLRP3 and pro-IL-1β expression by targeting NLRP3 and p65 and licensing NLRP3 inflammasome activation.

[1] Department of Immunology, Key Laboratory for Experimental Teratology of the Chinese Ministry of Education, School of Basic Medical Science, Cheeloo College of Medicine, Shandong University, 250012 Jinan, Shandong, China. [2] State Key Laboratory of Microbial Technology, Shandong University, 250012 Jinan, Shandong, China. [3] Department of Cell Biology, School of Basic Medical Science, Cheeloo College of Medicine, Shandong University, 250012 Jinan, Shandong, China. [4] These authors contributed equally: Hui Song, Chunyuan Zhao. ✉email: wzhao@sdu.edu.cn

T he NLRP3 inflammasome is a multimolecular complex, comprising a NOD-like receptor NLRP3, an adapter protein apoptosis-associated speck-like protein (ASC), and an effector pro-caspase-1, and plays fundamental roles in inflammation[1–4]. The NLRP3 inflammasome recognizes pathogen-associated molecular patterns (PAMPs) from invading microbes and endogenous danger signals (damage-associated molecular patterns, DAMPs) released from damaged or dying cells[4–6]. NLRP3 assembles an inflammasome complex with ASC and procaspase-1 after receiving a priming signal (signal 1) from toll-like receptors (TLRs) and an activation signal (signal 2) from respective NLRP3 inflammasome activators, such as extracellular ATP, nigericin (Nig), alum, crystals, amyloid-β, and others. The NLRP3 inflammasome complex subsequently serves as a platform for self-cleavage and activation of the cysteine protease caspase-1, promotes the maturation and secretion of IL-1β and IL-18, and induces pyroptosis. Aberrant NLRP3 inflammasome activation is involved in many types of diseases, such as infectious diseases, gout, type 2 diabetes, atherosclerosis, Alzheimer's disease, and cancers. Therefore, NLRP3 inflammasome activity should be tightly regulated to avoid such detrimental disorders.

Ubiquitination is a key post-translational modification (PTM) that controls NLRP3 inflammasome activation[7,8]. In resting macrophages, NLRP3 is polyubiquitinated with mixed K48 and K63 ubiquitin chains, which is crucial for the maintenance of NLRP3 inactivation. NLRP3 is deubiquitylated upon priming and activation, and that is a key step in the formation and activation of the NLRP3 inflammasome[9,10]. ABRO1 recruits BRCC3 to remove K63-linked ubiquitination of NLRP3, facilitating the assembly and activation of NLRP3 inflammasome[11]. K48-linked ubiquitination mediates protein degradation of NLRP3 and thus limits NLRP3 inflammasome activation. Although several E3 ubiquitin ligases such as TRIM31, March7, ARIH2, and FBXL2[9,12–14] have been reported to attenuate NLRP3 inflammasome activation by mediating NLRP3 protein degradation, the function of K48-linked deubiquitination on NLRP3 inflammasome activity remains largely unclear. Whether any deubiquitinating enzymes exist to specifically remove K48-linked ubiquitination of NLRP3, stabilize its expression, and thus license NLRP3 inflammasome activation, remains to be investigated.

Ubiquitin specific peptidase 1 (USP1)-associated factor 1 (UAF1, also called WDR48 or p80) is a stoichiometric binding partner of three deubiquitinating enzymes and constitutes three deubiquitinating enzyme complexes, including UAF1/USP1, UAF1/USP12, and UAF1/USP46. UAF1 constitutively binds to USP1, USP12, and USP46, and this binding greatly enhances their deubiquitinase activity. The UAF1/USP1 complex deubiquitinates a wide range of substrates and has been implicated in the regulation of DNA repair processes[15–18], tumor pathogenesis[19–23], and antiviral innate immunity[24]. USP1 can deubiquitinate and stabilize inhibitors of DNA binding proteins (IDs) and subsequently promote the maintenance of mesenchymal stem cells in osteosarcoma[6,15]. USP12 and USP46 are also involved in tumor pathogenesis by deubiquitinating and stabilizing different targets, such as PH domain leucine-rich repeat protein phosphatase 1 (PHLPP1), TP53 and androgen receptor (AR)[24–28]. USP12 deubiquitylates and prevents lysosomal degradation of LAT and Trat1 to maintain the proximal T-cell receptor (TCR) complex for the duration of signaling[29]. However, the potential roles of UAF1 deubiquitinase complexes in inflammation are unclear.

Here, we show that UAF1 facilitates NLRP3 inflammasome activation by recruiting the deubiquitinases USP1, USP12, and USP46. The UAF1/USP1 complex interacts with NLRP3, removes its K48-linked polyubiquitination, and stabilizes NLRP3 protein. The UAF1/USP12 and UAF1/USP46 complexes interact with p65 and inhibit its ubiquitination and degradation, thus promoting

NF-κB activation, resulting in the enhancements of NLRP3 and pro-IL-1β expressions. Consequently, the UAF1 deubiquitinase complexes facilitates NLRP3 inflammasome activation via targeting NLRP3 and p65. Our study thus uncovers mechanisms regulating NLRP3 inflammasome activation and suggests a promising approach for modulating NLRP3-dependent immunopathologies.

## Results

**UAF1 facilitates NLRP3 inflammasome activation**. To investigate the potential role of UAF1 in NLRP3 inflammasome activation, we first examined the effects of *Uaf1* deficiency on IL-1β secretion and caspase-1 cleavage. Systemic deletion of *Uaf1* causes early embryonic lethality in mice[30]. Therefore, we generated *Uaf1*^flox/flox mice and then crossed them with *Lyz2*-Cre mice to specifically knockout *Uaf1* in myeloid cells (called '*Uaf1*^CKO' here) (Supplementary Fig. 1). ATP-stimulated and Nig-stimulated IL-1β secretion and caspase-1 cleavage in LPS-primed *Uaf1*-deficient mouse peritoneal macrophages were markedly attenuated (Fig. 1a–c). In addition, *Uaf1* deficiency could inhibit TNF and IL-6 secretions. To confirm the intrinsic role of UAF1, siRNAs targeting mouse *Uaf1* were used to suppress endogenous UAF1 expression (Supplementary Fig. 2a). *Uaf1* knockdown substantially suppressed NLRP3 inflammasome activation-dependent IL-1β secretion and caspase-1 cleavage in mouse peritoneal macrophages (Supplementary Fig. 2b, c). We next investigated the physiological relevance of the effects of UAF1 on NLRP3 inflammasome activation in vivo. Induction of IL-1β by intraperitoneal (i.p.) injection of LPS was shown to be NLRP3 dependent[31,32]. In our study, IL-1β secretion induced by the LPS injection was much lower in the sera of *Uaf1*^CKO mice than in the sera of WT mice, indicating that *Uaf1* deficiency inhibits NLRP3 inflammasome activation in vivo (Fig. 1d). Furthermore, TNF and IL-6 secretions were also reduced in the sera of *Uaf1*^CKO mice compared with WT mice. Collectively, these results indicate that UAF1 facilitates NLRP3 inflammasome activation, and promotes TNF and IL-6 secretions.

**UAF1 enhances NLRP3 expression**. Cellular NLRP3 level is vital for the assembly and activation of NLRP3 inflammasome. We then examined the effects of UAF1 on NLRP3 expression. Deficiency and knockdown of *Uaf1* markedly inhibited NLRP3 protein expression (Figs. 1b, c and 2a and Supplementary Fig. 2c). In addition, *Nlrp3* mRNA expression was also considerably attenuated by *Uaf1* deficiency (Fig. 2b). These data indicate that UAF1 selectively controls NLRP3 expression at both the protein and mRNA levels. *Uaf1* deficiency had no effect on the expression of AIM2 and NLRC4, other two pattern recognition receptors (PRRs), which could form inflammasomes (Fig. 2c). However, IL-1β secretion induced by AIM2 and NLRC4 inflammasome activation was decreased in *Uaf1* deficiency (Supplementary Fig. 3a). The evidence promotes us to examine the potential roles of UAF1 on NF-κB activation, a key transcription factor controlling *Nlrp3*, *Il1b*, *Tnf*, and *Il6* transcription. *Uaf1* deficiency inhibited LPS-induced and poly(I:C)-induced phosphorylation of p65 (Fig. 2d, e). LPS-induced TNF and IL-6 expression was attenuated by *Uaf1* deficiency (Fig. 1a and Supplementary Fig. 3b). Moreover, *Uaf1* deficiency inhibited LPS-induced Il1b mRNA expression (Supplementary Fig. 3c). Furthermore, UAF1 overexpression enhanced MyD88-induced and TRIF-induced NF-κB activation (Fig. 2f). Taken together, these data show that UAF1 enhances NLRP3 and pro-IL-1β expression.

**USP1/12/46 enhances NLRP3 expression and inflammasome activation**. To investigate whether the regulatory role of UAF1 on

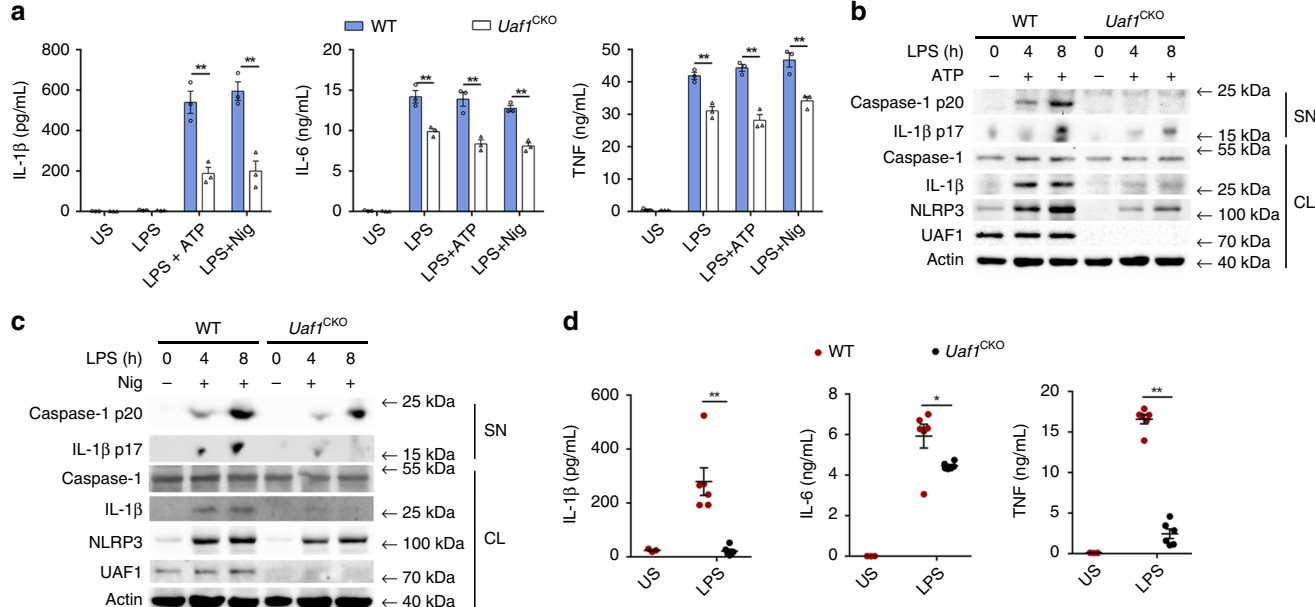

**Fig. 1 UAF1 facilitates NLRP3 inflammasome activation. a** ELISA analysis of IL-1β, IL-6 and TNF secretion in mouse peritoneal macrophages from wild-type (WT) or *Uaf1*CKO mice following priming with LPS for 8 h and subsequent stimulation with ATP or Nig for 40 min. US, unstimulated (mean ± SEM, two-tailed *t*-test *Uaf1*CKO vs. WT, left panel: **P = 0.0050, 0.0044 in sequence, middle panel: **P = 0.0070, 0.0052, 0.0009 in sequence, right panel: **P = 0.0035, 0.0014, 0.0079 in sequence; n = 3 independent experiments). **b, c** Western blot analysis of caspase-1 and IL-1β cleavage in mouse peritoneal macrophages from WT or *Uaf1*CKO mice after priming with LPS for the indicated time periods and then stimulated with ATP (**b**) or Nig (**c**) for 40 min. **d** ELISA analysis of serum levels of IL-1β, IL-6 and TNF from WT or *Uaf1*CKO mice after i.p. injection of LPS for 90 min (mean ± SEM, two-tailed *t*-test *Uaf1*CKO vs. WT, left panel: **P = 0.0005, middle panel: *P = 0.0321, right panel: **P < 0.0001; n = 6 independent experiments). US, unstimulated; SN, supernatants; CL, cell lysates. Similar results were obtained from three independent experiments.

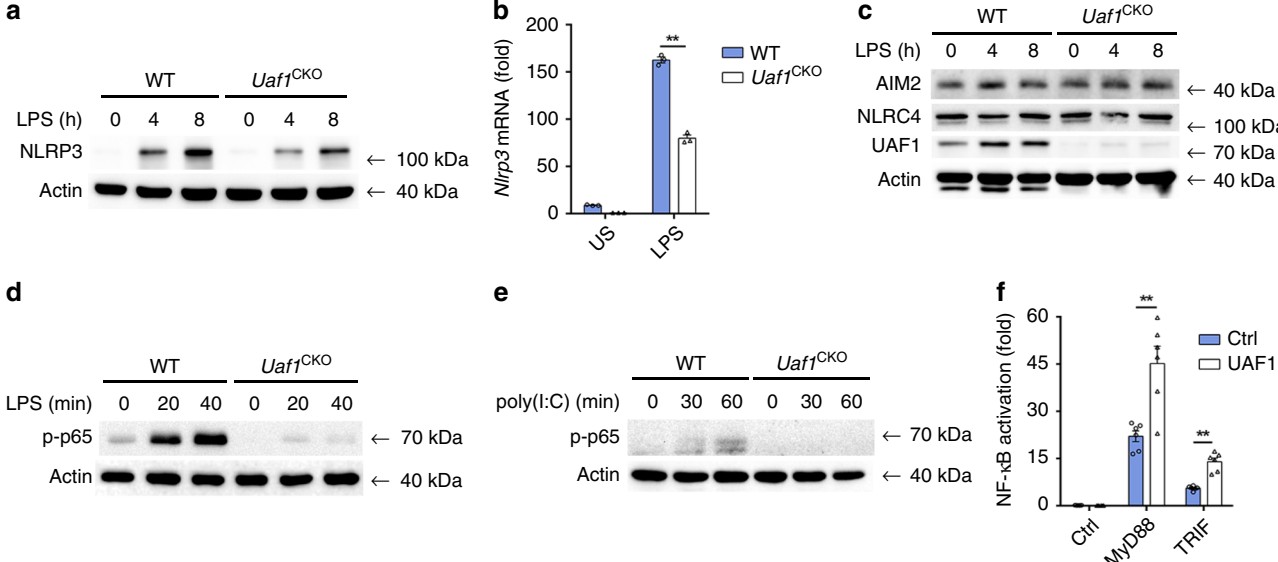

**Fig. 2 UAF1 enhances NLRP3 expression. a, c** Western blot analysis of NLRP3 (**a**), AIM2 and NLRC4 (**c**) in WT or *Uaf1*-deficient mouse peritoneal macrophages stimulated with LPS for the indicated time periods. **b** RT-PCR analysis of *Nlrp3* mRNA expression in WT or *Uaf1*-deficient mouse peritoneal macrophages stimulated with LPS for 2 h (mean ± SEM, two-tailed *t*-test *Uaf1*CKO vs. WT, **P < 0.0001; n = 3 independent experiments). **d, e** Western blot analysis of p-p65 in WT or *Uaf1*-deficient mouse peritoneal macrophages stimulated with LPS (**d**) or poly(I:C) (**e**) for the indicated time periods. **f** Analysis of NF-κB reporter activation using a Luciferase assay in HEK293T cells transiently transfected with NF-κB reporter plasmid together with MyD88 or TRIF, and UAF1 expression plasmid or empty control plasmid (mean ± SEM, two-tailed *t*-test UAF1 plasmid vs. empty control plasmid, **P = 0.0024, <0.0001 in sequence; n = 6 independent experiments). Similar results were obtained from three independent experiments.

NLRP3 expression was ubiquitination-dependent, we examined the effects of USP1, USP12, and USP46 on NLRP3 expression and NLRP3 inflammasome activation. SiRNAs were used to suppress endogenous USP1, USP12, or USP46 expression. *Usp1* siRNA 2,

*Usp12* siRNA 3, and *Usp46* siRNA 1, which have higher efficiencies to inhibit the expression of respective targets (Supplementary Fig. 4a–c), were used in the following experiments. *Usp1*, *Usp12*, or *Usp46* knockdown considerably inhibited IL-1β

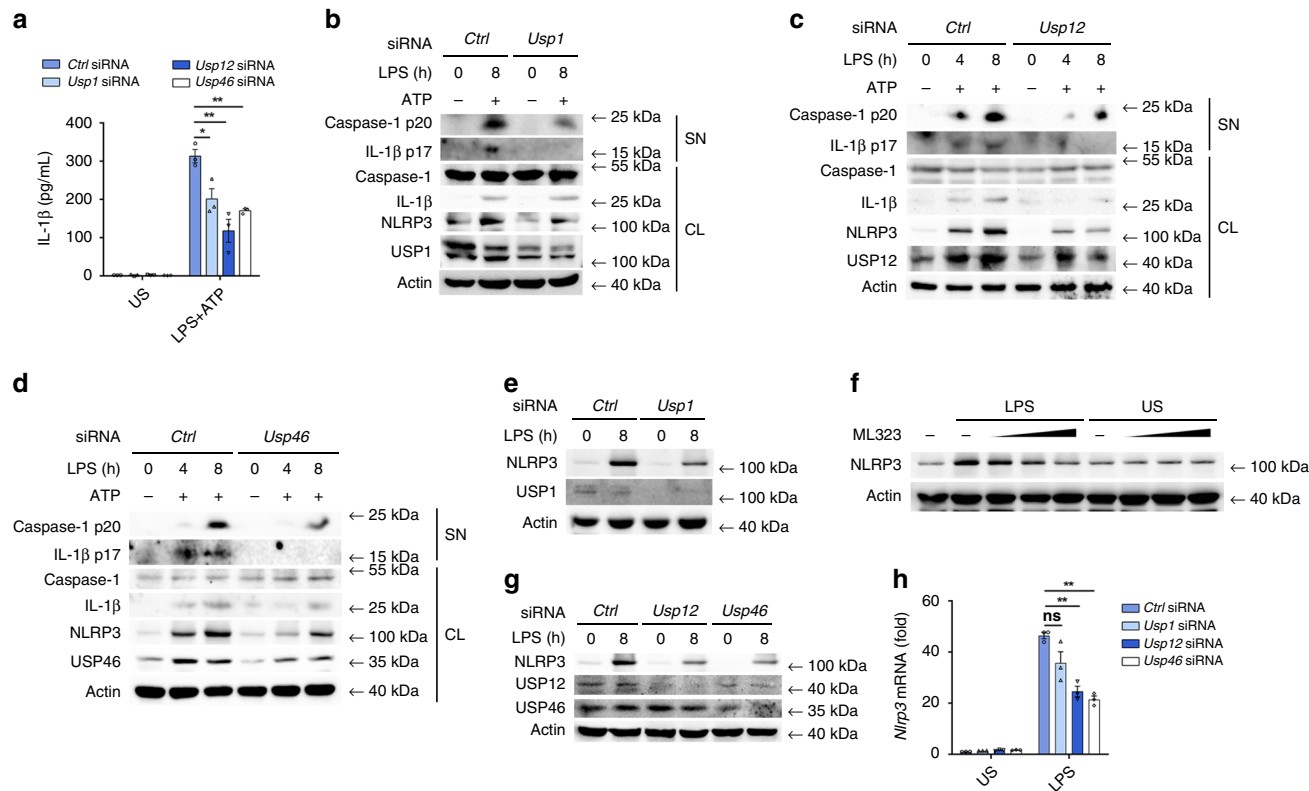

**Fig. 3 USP1, USP12, and USP46 enhance NLRP3 inflammasome activation. a–d** ELISA analysis of IL-1β secretion (**a**) and western blot analysis of caspase-1 and IL-1β cleavage (**b, d**) in *Ctrl* siRNA-, *Usp1* siRNA-, *Usp12* siRNA-, or *Usp46* siRNA-transfected mouse peritoneal macrophages followed by priming with LPS for 8 h (**a**) or indicated time periods (**b–d**), and then stimulated with ATP for 40 min (mean ± SEM, two-tailed *t*-test *Usp1* siRNA, *Usp12* siRNA and *Usp46* siRNA vs. *Ctrl* siRNA, *\*P* = 0.0238, *\*\*P* = 0.0049, 0.0013 in sequence; *n* = 3 independent experiments). **e, g** Western blot analysis of NLRP3 expression in *Ctrl* siRNA-, *Usp1* siRNA-, *Usp12* siRNA-, or *Usp46* siRNA-transfected mouse peritoneal macrophages after stimulation with LPS for 8 h. **f** Western blot analysis of NLRP3 expression in mouse peritoneal macrophages treated with DMSO or increasing amount of ML323 (0, 15, 30, and 60 μM) followed by stimulation with LPS for 8 h. **h** RT-PCR analysis of *Nlrp3* mRNA expression in *Ctrl* siRNA-, *Usp1* siRNA-, *Usp12* siRNA-, or *Usp46* siRNA-transfected mouse peritoneal macrophages, followed by stimulation with LPS for 2 h (mean ± SEM, two-tailed *t*-test *Usp1* siRNA, *Usp12* siRNA, and *Usp46* siRNA vs. *Ctrl* siRNA, ns = 0.0813, *\*\*P* = 0.0008, 0.0002 in sequence; *n* = 3 independent experiments). US, unstimulated; SN, supernatants; CL, cell lysates. Similar results were obtained from three independent experiments.

secretion, caspase-1 and IL-1β cleavage in LPS-primed and ATP-stimulated macrophages (Fig. 3a–d), indicating that USP1, USP12, and USP46 promoted NLRP3 inflammasome activation.

Although *Usp1* knockdown had no effect on NLRP3 expression in unstimulated macrophages, it markedly attenuated LPS-induced NLRP3 expression (Fig. 3e). Furthermore, ML323, an allosteric and specific inhibitor of the UAF1/USP1 deubiquitinase complex[33], inhibited LPS-induced NLRP3 expression in a dose-dependent manner (Fig. 3f). However, both ML323 and *Usp1* knockdown did not affect the expression of AIM2 and NLRC4 (Supplementary Fig. 4d, e). Interestingly, *Usp12* and *Usp46* knockdown markedly inhibited NLRP3 expression in both unstimulated and LPS-stimulated macrophages (Fig. 3g). In addition, both *Usp12* and *Usp46* knockdown significantly inhibited *Nlrp3* mRNA expression; however, *Usp1* knockdown had no effect on *Nlrp3* mRNA expression (Fig. 3h). Collectively, these data indicate that USP1, USP12, and USP46 suppressed NLRP3 expression via different mechanisms.

**UAF1/USP12 and UAF1/USP46 complexes promote NLRP3 transcription.** Both *Usp12* and *Usp46* knockdown suppressed LPS-induced *Il1b*, *Tnf*, *Il6*, and *Nlrp3* mRNA expression (Figs. 3h and 4a), suggesting that USP12 and USP46 may regulate the priming process of NLRP3 inflammasome activation. The transcription factor NF-κB is critical for NLRP3 expression[34]. We

thus examined the effects of USP12 and USP46 on NF-κB activation. Both USP12 and USP46 significantly enhanced MyD88-, TRAF6-, TAK1-, and p65-induced NF-κB luciferase reporter activation (Fig. 4b and Supplementary Fig. 5a), indicating that USP12 and USP46 promoted NF-κB activation by targeting p65 or downstream of p65. Moreover, both *Usp12* and *Usp46* knockdown considerably attenuated LPS-induced phosphorylation of p65 and total p65 expression (Fig. 4c, d), suggesting that p65 may be the target of USP12 and USP46. As parallel controls, USP1 overexpression had no effect on MyD88, TRAF6, TAK1, and p65-induced NF-κB luciferase reporter activation (Fig. 4b and Supplementary Fig. 5a). In addition, no difference in p65 expression was observed in *Usp1* siRNA transfected macrophages (Supplementary Fig. 5b). Furthermore, *Uaf1* deficiency also markedly inhibited p65 expression and phosphorylation of p65 (Figs. 2d, 4e). Collectively, these results indicate that UAF1/USP12 and UAF1/USP46 complexes promote NF-κB activation by enhancing p65 expression.

NF-κB subunit p65 can be inactivated and degraded by proteasomes[35]. We then examined whether USP12/USP46 could target p65 to remove its polyubiquitin chains. P65 interacted with USP12, USP46 and UAF1 in both resting and LPS-stimulated macrophages, but exhibited no association with USP1 (Fig. 4f). Consistently, the interaction between exogenously expressed p65 and USP12/USP46/UAF1 was observed in HEK293T cells

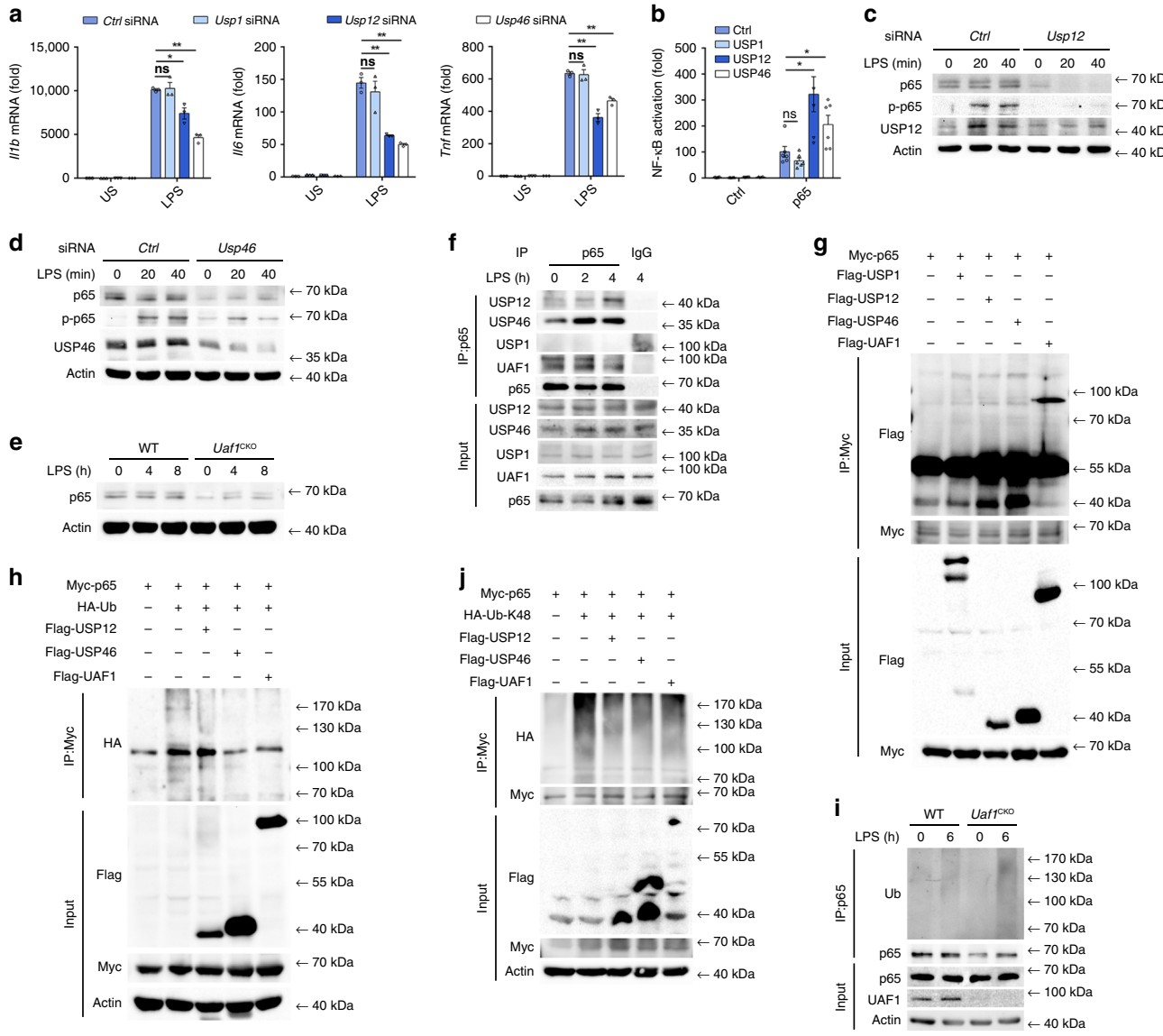

**Fig. 4 UAF1/USP12 and UAF1/USP46 complexes promote NLRP3 transcription. a** RT-PCR analysis of *Il1b*, *Il6* and *Tnf* mRNA expression in *Ctrl* siRNA−, *Usp1* siRNA−, *Usp12* siRNA−, or *Usp46* siRNA-transfected mouse peritoneal macrophages, followed by stimulation with LPS for 2 h (mean ± SEM, two-tailed t-test *Usp1* siRNA, *Usp12* siRNA and *Usp46* siRNA vs. *Ctrl* siRNA, left panel: ns = 0.8310, *P = 0.0135, **P < 0.0001 in sequence, middle panel: ns = 0.4945, **P = 0.0006, 0.0003 in sequence, right panel: ns = 0.8037, **P = 0.0005, 0.0007 in sequence; n = 3 independent experiments). **b** Analysis of NF-κB reporter activation using a Luciferase assay in HEK293T cells transiently transfected with the indicated plasmids (mean ± SEM, two-tailed *t*-test USP1, USP12 and USP46 plasmids vs. empty control plasmid, ns = 0.1774, *P = 0.0106, 0.0298 in sequence; n = 6 independent experiments). **c, d** Western blot analysis of p65 or p-p65 in *Ctrl* siRNA-, *Usp12* siRNA-, or *Usp46* siRNA-transfected mouse peritoneal macrophages followed by stimulation with LPS for the indicated time periods. **e** Western blot analysis of p65 in WT or *Uaf1* deficient mouse peritoneal macrophages after stimulation with LPS for the indicated time periods. **f** Cell lysates of mouse peritoneal macrophages stimulated with LPS for the indicated time periods were subjected to immunoprecipitation (IP) with anti-p65, followed by western blot analysis with antibodies to UAF1, USP1, USP12, USP46, or p65. Proteins from a whole-cell lysate were used as a positive control (Input). **g** Cell lysates of HEK293T cells transiently transfected with Myc-p65, Flag-USP1, Flag-USP12, Flag-USP46, and Flag-UAF1 were subjected to IP using Myc antibody, followed by western blot analysis with anti-Myc or anti-Flag. **h, j** Cell lysates of HEK293T cells transiently transfected with Myc-p65, HA-Ub, or HA-tagged K48-linked ubiquitin (HA-K48-Ub), Flag-USP12, USP46, and Flag-UAF1 were subjected to IP with Myc antibody, followed by western blot analysis with anti-HA. **i** Western blot analysis of extracts from LPS-stimulated WT or *Uaf1*CKO macrophages, followed by IP with anti-p65, and subsequent probing with anti-Ub. US, unstimulated. Similar results were obtained from three independent experiments.

(Fig. 4g). We then examined the effects of USP12/USP46/UAF1 on p65 polyubiquitination and observed that all the three could remove its polyubiquitination (Fig. 4h). Under physiological conditions, endogenous p65 was ubiquitinated upon LPS stimulation in macrophages, and *Uaf1* deficiency markedly increased endogenous p65 ubiquitination (Fig. 4i). To study the forms of USP12/USP46/UAF1-mediated deubiquitination of p65, ubiquitin mutant vectors K48 and K63, in which all their lysine residues were substituted with arginine except at positions 48 and 63, respectively, were used in the transfection assays. UAF1, USP12, and USP46 markedly inhibited K48-linked ubiquitination of p65 (Fig. 4j), but had no effect on K63-linked ubiquitination of p65 (Supplementary Fig. 5d). Taken together, these results indicate that the UAF1/USP12 and UAF1/USP46 complexes remove K48-linked ubiquitination of p65 and enhance its expression, thereby promoting NLRP3 transcription.

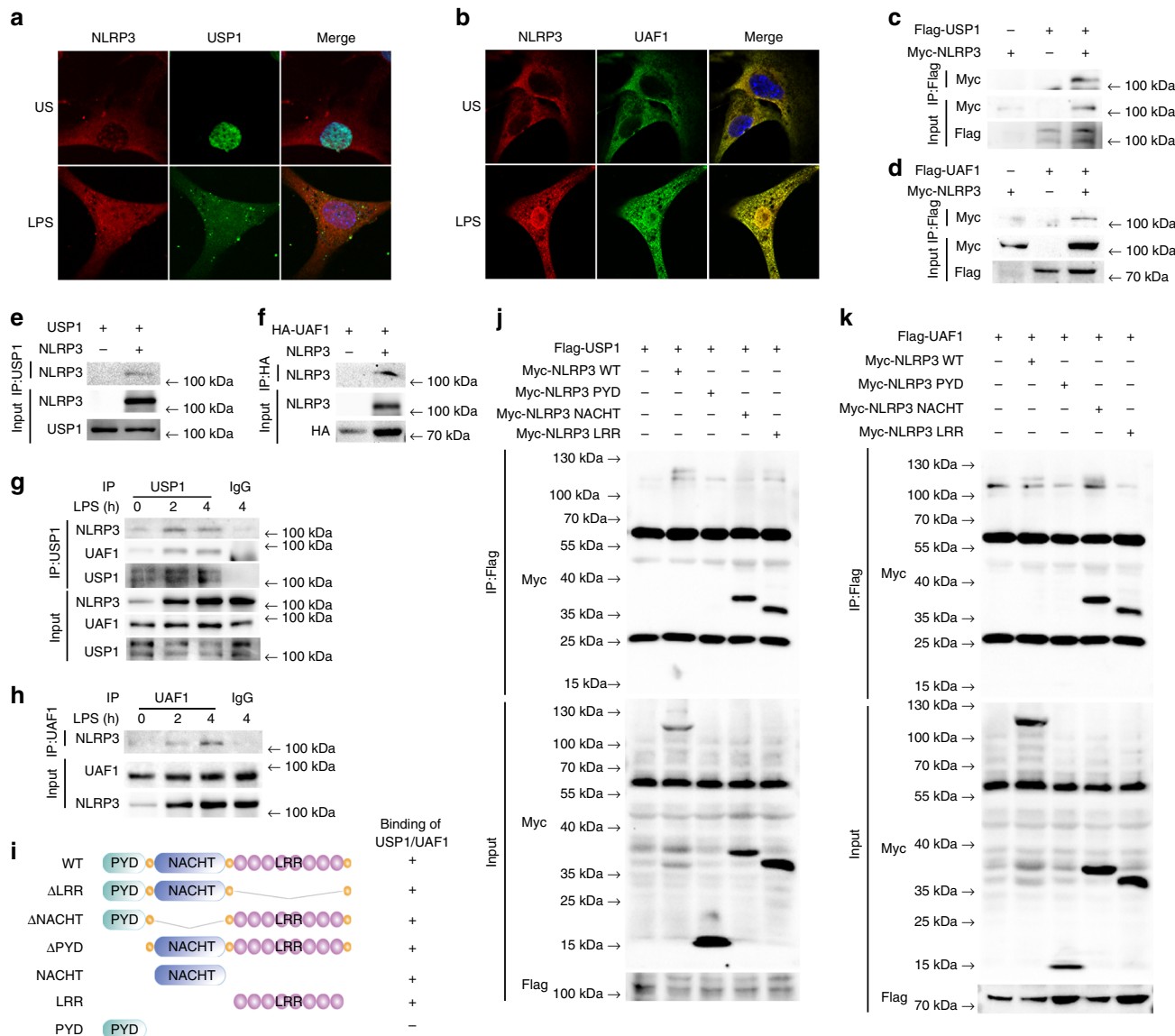

**Fig. 5 UAF1/USP1 interacts with NLRP3. a, b** Confocal microscopy analysis of colocalization of NLRP3 with USP1 or UAF1. MEFs transfected with Myc-NLRP3, GFP-USP1 or Flag-UAF1 were stimulated with LPS for 2 h, then fixed and incubated with a secondary antibody conjugated to Alexa Fluor 637 or Alexa Fluor 488. Scale bar, 10 μm. **c, d** Extracts from HEK293T cells transiently transfected with Flag-USP1 and Myc-NLRP3 (**c**), and Flag-UAF1 and Myc-NLRP3 (**d**) were subjected to IP with anti-Flag, followed by western blot analysis with anti-Myc and anti-Flag. **e** PCMV6-USP1 and Myc-tagged NLRP3 were obtained by in vitro transcription and translation. Interaction between USP1 and NLRP3 was analyzed by mixing USP1 and NLRP3 together, followed by IP with USP1 antibody and immunoblot analysis with NLRP3 and USP1 antibody. **f** HA-tagged UAF1 and Myc-tagged NLRP3 were obtained by in vitro transcription and translation. Interaction between UAF1 and NLRP3 was assayed by mixing UAF1 and NLRP3 together, followed by IP with HA antibody and immunoblot analysis with NLRP3 and HA antibody. **g, h** Extracts of peritoneal macrophages stimulated with LPS for the indicated time periods were subjected to immunoprecipitation with anti-USP1 (**g**) and anti-UAF1 (**h**) followed by western blot analysis with indicated antibody. Proteins from a whole-cell lysate were used as positive control. **i** Schematic diagram of NLRP3 and its truncation mutants. **j, k** Myc-tagged NLRP3 or its mutants along with Flag-USP1 (**j**) or Flag-UAF1 (**k**) were individually transfected into HEK293T cells. The cell lysates were immunoprecipitated with Flag antibody and then immunoblotted with the indicated antibodies. US, unstimulated. Similar results were obtained from three independent experiments.

**UAF1/USP1 interacts with NLRP3.** USP1 enhanced NLRP3 protein expression, with no effect on *Nlrp3* mRNA expression, suggesting that the UAF1/USP1 complex regulated posttranslational modification of NLRP3. We first examined the association between NLRP3 and UAF1/USP1. Confocal analysis demonstrated the colocalization between USP1/UAF1 and NLRP3 upon LPS stimulation (Fig. 5a, b). We expressed NLRP3 along with USP1 and UAF1 in HEK293T cells. NLRP3 could be immunoprecipitated with USP1 and UAF1 (Fig. 5c, d). In vitro binding assays demonstrated that NLRP3 could directly interact with USP1 and UAF1 (Fig. 5e, f). Next, to confirm the interaction

between NLRP3 and UAF1/USP1 under physiological conditions, we assessed resting and LPS-stimulated macrophages by immunoprecipitation. An association between NLRP3 and USP1/UAF1 was detected in LPS-stimulated mouse peritoneal macrophages (Fig. 5g, h). Taken together, these data suggest that NLRP3 could directly interact with UAF1/USP1.

To search for the domains of NLRP3 responsible for the interaction with USP1 and UAF1, a series of Myc-tagged NLRP3 truncated mutants were constructed (Fig. 5i). USP1 and UAF1 were coprecipitated with NLRP3 wild-type (WT), LRR domain deletion mutant (ΔLRR), NACHT domain deletion mutant

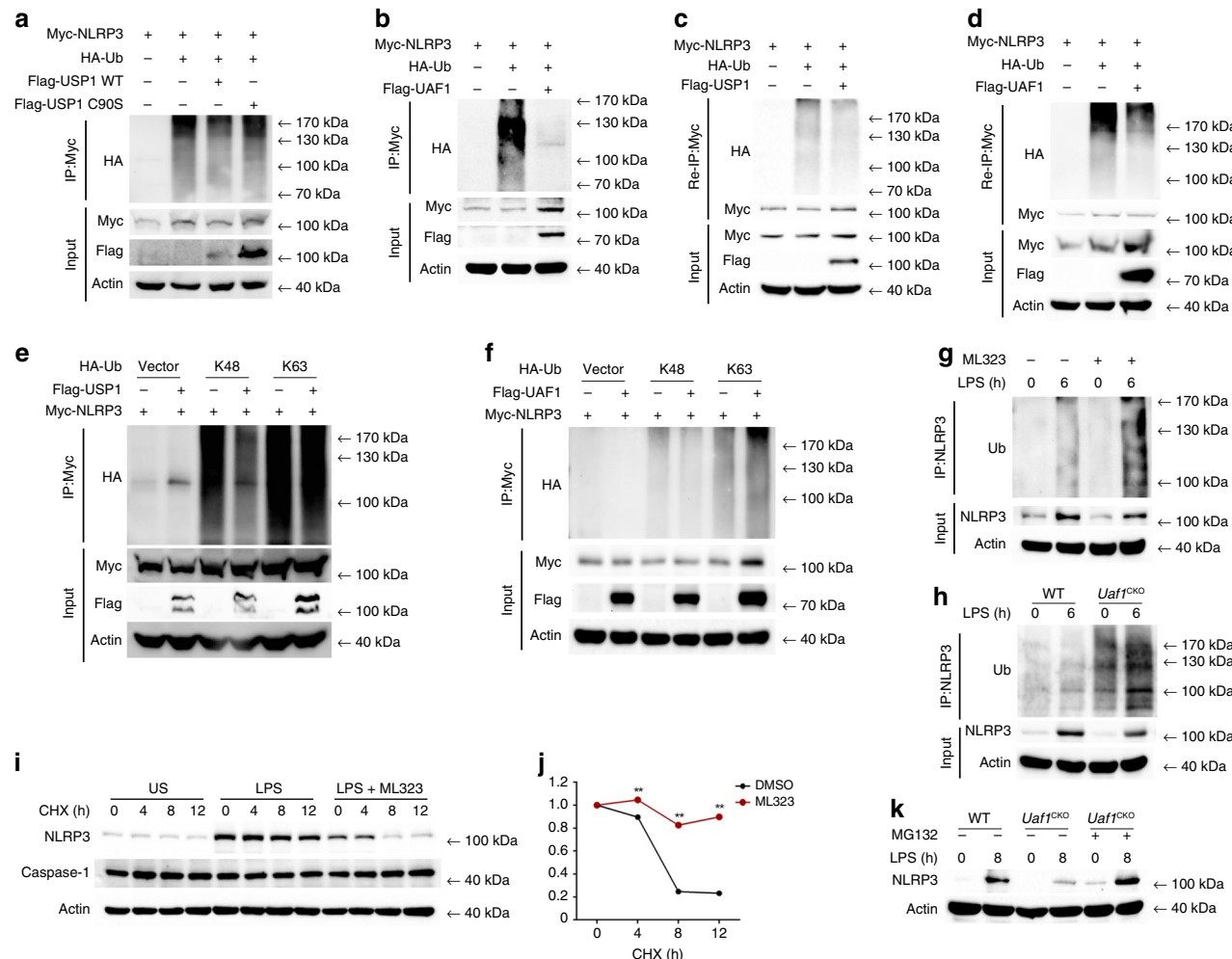

**Fig. 6 UAF1/USP1 removes NLRP3 ubiquitination and stabilizes NLRP3. a, b** Western blot analysis of extracts obtained from HEK293T cells transfected with HA-Ub, Myc-NLRP3, Flag-USP1 WT, and USP1 C90S (**a**) or Flag-UAF1 (**b**), followed by IP with anti-Myc, and then probed with anti-HA. **c, d** Extracts obtained from HEK293T cells transiently cotransfected with NLRP3-Myc, HA-Ub, and Flag-USP1 (**c**) or Flag-UAF1 (**d**) were immunoprecipitated using Myc antibody. The immunoprecipitates were denatured and reimmunoprecipitated with Myc antibody (two-step immunoprecipitation, Re-IP), and then immunoblotted with the indicated antibodies. **e, f** Western blot analysis of extracts from HEK293T cells transfected with HA-K48-Ub or HA-K63-Ub, Myc-NLRP3 and Flag-USP1 (**e**) or Flag-UAF1 (**f**), followed by IP with anti-Myc and subsequently probed with anti-HA. **g** MEFs were pretreated with DMSO or ML323 for 4 h and then stimulated with LPS for 6 h. Cell lysates were immunoprecipitated with anti-NLRP3, followed by western blot analysis with anti-Ub. **h** Cell lysates from WT or *Uaf1* deficient mouse peritoneal macrophages stimulated with LPS were immunoprecipitated with anti-NLRP3, followed by western blot analysis with anti-Ub. **i, j** Western blot analysis of NLRP3 and caspase-1 expression in mouse peritoneal macrophages pretreated with DMSO or ML323 for 4 h and then stimulated with LPS for 4 h, followed by treatment with cycloheximide (CHX) for the indicated time periods. NLRP3 expression level was quantitated by measuring band intensities using 'ImageJ' software (**j**). The values were normalized to actin (mean ± SEM, two-tailed *t*-test ML323 vs. DMSO, \*\*$P < 0.0001$, respectively; $n = 3$ independent experiments). **k** Western blot analysis of NLRP3 expression in WT and *Uaf1* deficient mouse peritoneal macrophages pretreated with LPS for 8 h, followed by treatment with MG132 for 4 h before harvesting the cells. Similar results were obtained from three independent experiments.

(ΔNACHT), PYD domain deletion mutant (ΔPYD), LRR domain mutant (LRR) and NACHT domain mutant (NACHT) (Fig. 5j, k and Supplementary Fig. 6). However, the PYD domain mutant (PYD) lost the ability to bind USP1 and UAF1 (Fig. 5j, k). These results indicate that the LRR and NACHT domains of NLRP3 interact with UAF1/USP1.

**UAF1/USP1 removes NLRP3 ubiquitination and stabilizes NLRP3.** NLRP3 could be ubiquitinated with both K48 and K63 linkage. We identified the deubiquitinase complex UAF1/USP1 as NLRP3-associated proteins (Fig. 5), which prompted us to investigate whether the complex could remove the ubiquitination of NLRP3. Both USP1 and UAF1 considerably inhibited ubiquitination of NLRP3 (Fig. 6a, b). However, the USP1 point

mutation (C90S) with substitution of the cysteine residue with serine at position 90, lost the ability to inhibit the poly-ubiquitination of NLRP3 (Fig. 6a), indicating that USP1 could remove the ubiquitination of NLRP3 through its deubiquitinase activity. Given that NLRP3 possibly forms a complex with other proteins, we performed a two-step immunoprecipitation assay (Re-IP) to exclude the effects of other proteins in the complex. In HEK293T cells, Myc-tagged NLRP3 was cotransfected with ubiquitin and USP1 or UAF1, respectively. The cell lysates were subjected to immunoprecipitation with anti-Myc, and then the immunoprecipitates were denatured, followed by reimmuno-cipitation with Myc antibody. Notably, NLRP3 polyubiquitination was markedly attenuated in the presence of USP1 or UAF1 (Fig. 6c, d). Furthermore, USP1 inhibited NLRP3

polyubiquitination with K48-linkage, but not with K63-linkage (Fig. 6e), indicating that USP1 selectively removed K48-linked polyubiquitination of NLRP3. Consistently, UAF1 also inhibited K48-linked polyubiquitination of NLRP3 (Fig. 6f). Under physiological conditions, endogenous NLRP3 was ubiquitinated upon LPS stimulation. Moreover, polyubiquitination of NLRP3 was enhanced in ML323-treated mouse embryonic fibroblasts (MEFs) (Fig. 6g) and *Uaf1*-deficient macrophages (Fig. 6h). Collectively, these data indicate that UAF1/USP1 directly eliminates K48-linked polyubiquitination of NLRP3 through its deubiquitinase activity.

K48-linked protein ubiquitination leads to degradation of the corresponding proteins. Both UAF1 and USP1 enhanced LPS-induced NLRP3 expression. We then investigated the effects of UAF1/USP1 complex on NLRP3 protein degradation by cycloheximide (CHX) chase experiment. ML323 treatment significantly promoted NLRP3 protein degradation, with no effect on caspase-1 (Fig. 6i, j). In addition, *Uaf1* deficiency resulted in a loss of the inhibitory effects on NLRP3 expression in MG132-treated macrophages (Fig. 6k). Taken together, these data indicate that the USP1/UAF1 deubiquitinase complex targets NLRP3 and inhibits its degradation, thus facilitating NLRP3 inflammasome activation.

**ML323 and *Uaf1* deficiency ameliorate NLRP3-dependent inflammation**. Next, we investigated whether ML323, a selective UAF1/USP1 inhibitor, could suppress NLRP3 inflammasome activation. ML323 treatment attenuated NLRP3-dependent IL-1β secretion and caspase-1 cleavage in macrophages (Fig. 7a, b). ML323 is known to be active in vivo[36]. We then examined the physiological relevance of the UAF1/USP1 complex in inflammation in vivo using ML323. ML323-treated mice produced significantly less IL-1β and TNF in sera after i.p. injection of LPS than control mice, while no difference in IL-6 secretion was observed (Fig. 7c).

Folic acid (FA)-induced acute tubular necrosis (ATN) is tightly associated with NLRP3 inflammasome activation[37]. To evaluate the roles of ML323 in FA-induced ATN, mice were i.p. injected with ML323 followed by an FA injection. ML323 treatment considerably reduced IL-1β secretion in sera (Fig. 7d) and NLRP3 expression in the kidneys (Fig. 7e). Renal inflammation and edema were greatly ameliorated in the kidneys of ML323-treated mice (Fig. 7f, g). Concordantly, ML323-treated mice were found to be more resistant in survival assays upon FA injection (Fig. 7h). Therefore, as a selective UAF1/USP1 complex inhibitor, ML323 inhibits IL-1β secretion and therefore ameliorates NLRP3-dependent inflammation both in vitro and in vivo.

We further investigated the physiological relevance of UAF1 on inflammation in FA-induced ATN using *Uaf1*[CKO] mice. *Uaf1* deficiency in myeloid cells markedly suppressed IL-1β secretion and pro-IL-1β expression in the kidneys, due to the regulatory roles of UAF1 in NF-κB activation and NLRP3 expression (Fig.7i); less severe renal inflammation and edema were observed in *Uaf1*[CKO] mice (Fig. 7j, k). These data indicate that *Uaf1* deficiency in myeloid cells ameliorates FA-induced ATN and suggest UAF1 as a physiological enhancer of inflammation.

## Discussion

The NLRP3 inflammasome is activated by a wide variety of stimuli, including PAMPs from bacteria, viruses, and fungi; endogenous DAMPs in sterile inflammation; and exposure to environmental irritants[1,7,38]. Thus, NLRP3 inflammasome activity is strongly associated with a variety of diseases; hence, its activation should be tightly controlled. In this study, we provide several lines of evidence to demonstrate that UAF1

deubiquitinase complexes promote NLRP3 inflammasome activation by enhancing NLRP3 expression. As a cofactor of several deubiquitinases, UAF1 plays different roles by combining with USP1, USP12, and USP46, respectively. On the one hand, UAF1/USP12 and UAF1/USP46 complexes stabilize p65 expression, thus promoting the NF-κB signaling pathway, which results in the enhancement of NLRP3, pro-IL-1β, TNF, and IL-6 transcription. On the other hand, UAF1/USP1 complex selectively interacts with NLRP3, removes NLRP3 K48-linked ubiquitination and prevents NLRP3 from proteasomal degradation. Therefore, UAF1 facilitates NLRP3 inflammasome activation by enhancing IL-1β and NLRP3 expression at the mRNA and protein levels. Previously, we reported that LPS induced UAF1 expression in macrophages[36]. Thus, UAF1 might be a feedback enhancer of NLRP3 inflammasome activation.

NLRP3 protein expression is considered to be a rate-limiting step and causes subsequent NLRP3 inflammasome activation. In resting macrophages, NLRP3 expression is relatively low; hence, NLRP3 inflammasome assembly is hardly induced. However, following stimulation with exogenous and endogenous factors, including TLR agonists and proinflammatory cytokines, NLRP3 expression is dramatically induced to establish quick responses to NLRP3 activators. At the transcriptional level, NF-κB is critical for *Nlrp3* mRNA expression[34]. Zinc finger protein A20 deubiquitinates NF-κB, thereby signaling molecules to suppress NF-κB activation and NLRP3 transcription[39]. Aryl hydrocarbon receptor (AhR) binds to the xenobiotic response elements (XREs) located within two NF-κB binding sites in the NLRP3 promoter and attenuates NLRP3 transcription and NLRP3 inflammasome activation[40]. In this study, we demonstrate that UAF1/USP12 and UAF1/USP46 complexes target p65 and promote NF-κB activation, resulting in the enhancement of NLRP3 transcription and NLRP3 inflammasome activation. In addition, the UAF1/USP12 and UAF1/USP46 complexes also enhance the transcription of proinflammatory cytokines, including pro-IL-1β, TNF, and IL-6, which further promote NLRP3 inflammasome activation. At posttranslational level, multiple E3 ubiquitin ligases promote NLRP3 ubiquitination and protein degradation, which limit NLRP3 inflammasome activity. In this study, we observed that the UAF1/USP1 deubiquitinase complex inhibits proteasomal degradation of NLRP3 and enhances NLRP3 inflammasome activation. Deubiquitination of NLRP3 is required for optimal NLRP3 inflammasome activation. Our results indicate that NLRP3 deubiquitination is important for the maintenance of NLRP3 protein level, thus providing a mechanism to elucidate the indispensable roles of ubiquitination in NLRP3 inflammasome activation.

UAF1, as a cofactor of USP1, USP12, and USP46, could enhance their deubiquitinase activity by forming stable UAF1/USP protein complexes[41]. UAF1/USP1, UAF1/USP12, and UAF1/USP46 complexes play vital roles in DNA repair processes and tumor pathogenesis. Here, we demonstrated that UAF1 works together with USP12/USP46 and USP1 to facilitate NLRP3 and pro-IL-1β expression, and license NLRP3 inflammasome activation. As NLRP3 inflammasome needs two signals (priming signal and activation signal) to be activated, it acts just like a precise lock. Under certain circumstances, UAF1 and its partners (USP1/USP12/USP46) work as multiple keys to unlock and tightly control the activation of NLRP3 inflammasome at different levels (Supplementary Fig. 7).

Inappropriate NLRP3 inflammasome activation has been implicated in a variety of diseases, which makes NLRP3 an attractive drug target. Many inhibitors of the NLRP3 pathway have been developed and extensively investigated in multiple NLRP3 inflammasome-related disease models. For example, MCC950, a best studied NLRP3 inhibitor, displays efficacy in a

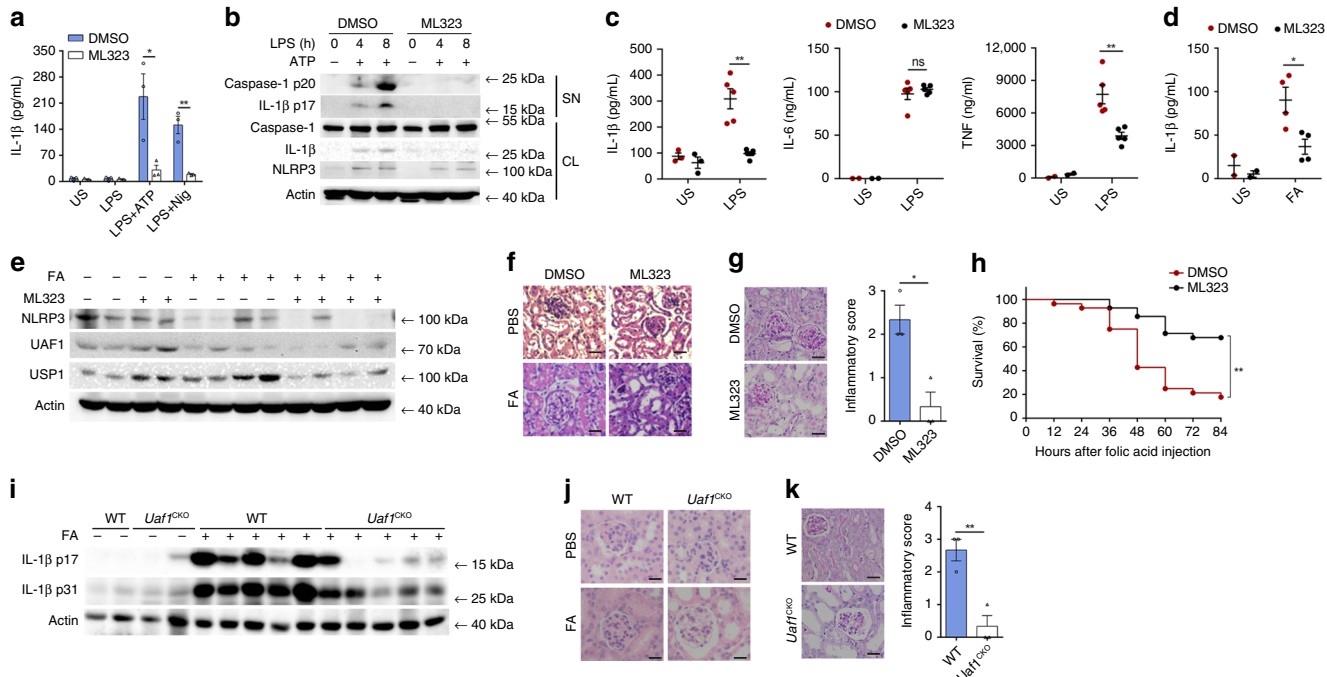

**Fig. 7 ML323 and *Uaf1* deficiency ameliorate NLRP3-dependent inflammation. a** ELISA analysis of IL-1β secretion in mouse peritoneal macrophages pretreated with DMSO or ML323 for 4 h, followed by LPS stimulation for 8 h and treatment with ATP or Nig for the last 40 min (mean ± SEM, two-tailed *t*-test ML323 vs. DMSO, *P = 0.0347, **P = 0.0049 in sequence; n = 3 independent experiments). **b** Western blot analysis of caspase-1 and IL-1β cleavage in mouse peritoneal macrophages pretreated with DMSO or ML323 for 4 h, followed by LPS stimulation for the indicated time periods and treatment with ATP for the last 40 min. **c** ELISA analysis of serum levels of IL-1β, IL-6, and TNF from C57BL/6 mice, initially i.p. injected with DMSO or ML323 for 4 h and later with LPS for 90 min (mean ± SEM, two-tailed t-test ML323 vs. DMSO, left panel: **P = 0.0007, middle panel: ns = 0.4847, right panel: **P = 0.0034; n = 5 independent experiments). **d** ELISA analysis of serum levels of IL-1β in C57BL/6 mice i.p. injected with DMSO or ML323 for 4 h and then i.p. injected with folic acid (FA) for 12 h (mean ± SEM, two-tailed t-test ML323 vs. DMSO, *P = 0.0193; n = 4 independent experiments). **e–g** C57BL/6 mice were i.p. injected with DMSO or ML323 for 4 h and then i.p. injected with folic acid (FA) for 36 h. Western blot analysis of NLRP3 expression in kidney samples (**e**), H&E staining of kidney tissue sections (**f**), PAS staining of kidney tissue sections and renal inflammation was scored based upon PAS staining (**g**) Scale bar, 20 μm (mean ± SEM, two-tailed *t*-test ML323 vs. DMSO, *P = 0.0132; n = 3 independent experiments). **h** Survival of the C57BL/6 mice i.p. injected with DMSO or ML323 for 4 h and then with FA (mean ± SEM, log rank test [Mantel-Cox] ML323 vs. DMSO, **P < 0.0001; n = 28 independent experiments). **i–k** WT or *Uaf1*CKO mice were i.p. injected with FA for 36 h. Western blot analysis of IL-1β p17 and p31 in kidney samples (**i**), H&E staining of kidney tissue sections (**j**), PAS staining of kidney tissue sections and renal inflammation was scored based upon PAS staining (**k**) (mean ± SEM, two-tailed t-test *Uaf1*CKO vs. WT, **P = 0.0078; n = 3 independent experiments). US, unstimulated; SN, supernatants; CL, cell lysates. Similar results were obtained from three independent experiments.

wide range of NLRP3-dependent murine disease models[32]. CY-09, another selective NLRP3 inhibitor, is also found to be efficacious in a mouse model of cryopyrin-associated periodic syndromes (CAPS)[42]. In this study, we identified ML323 as a novel NLRP3 inflammasome inhibitor. ML323 is a specific inhibitor of the UAF1/USP1 complex, with no inhibitory effects on both UAF1/USP12 and UAF1/USP46 complexes[33]. ML323 inhibits NLRP3 protein expression, suppresses NLRP3 inflammasome activation, thereby ameliorating NLRP3-dependent inflammation both in vitro and in vivo. Our results suggest that ML323 might be a promising candidate for the treatment of inflammatory diseases caused by aberrant NLRP3 activity.

In conclusion, by identifying UAF1 as a critical enhancer of NLRP3 and pro-IL-1β expression, this study provides insights into the mechanism of NLRP3 inflammasome activation. UAF1 enhances NLRP3 expression at both the mRNA and protein levels, as well as pro-IL-1β transcription through recruiting USP1/12/46. The UAF1/USP1 deubiquitinase complex directly targets NLRP3 and inhibits its ubiquitous degradation. The UAF1/USP12 and UAF1/USP46 complexes suppress the ubiquitous degradation of p65, promote NF-κB activation, and enhance NLRP3 and pro-IL-1β transcription. These results indicate that the UAF1 deubiquitinase complexes enhance NLRP3 and pro-IL-

1β expressions via targeting NLRP3 and p65, and thus facilitates NLRP3 inflammasome activation.

## Methods

**Mice.** *Uaf1*flox/flox mice on a C57BL/6J background were generated by Cyagen Biosciences Inc. (Guangzhou, China) using CRISPR-Pro technology. *Lyz2*-Cre mice were from Jackson Laboratory. *Uaf1*flox/flox mice were crossed with *Lyz2*-Cre transgenic mice to obtain *Uaf1*fl/fl*Lyz2*-Cre mice with *Uaf1* deficiency in myeloid cells, resulting in *Uaf1*-deficient mice (*Uaf1*CKO mice) with 984 missing base pairs (6350-7334, containing the third exon of *Uaf1* gene NM_026236.3), which caused a frameshift. For animal experiments with *Uaf1*CKO mice, littermate controls with normal *Uaf1* expression (*Uaf1*fl/fl) were used. C57BL/6J mice were obtained from Beijing Vital River Laboratory Animal Technology Co., Ltd. (Beijing, China). All animal experiments were performed in accordance with the National Institute of Health Guide for the Care and Use of Laboratory Animals; approval was obtained from the Scientific Investigation Board of Medical School of Shandong University (Jinan, Shandong Province, China). All the animals were generated in specific pathogen-free (SPF) levels with 40–70% humidity and daily cycles of 12 h of light at 23 °C and 12 h of dark at 21 °C.

**Reagents.** ATP, MG132, LPS (Escherichia coli, 055:B5), poly(I:C), folic acid, anti-HA (HA-7, cat. H3663, 1:1000 for WB), anti-Myc (9E10, cat. M4439, 1:1000 for WB), and anti-Flag (M2, cat. F1804, 1:1000 for WB, 1:400 for IP) were purchased from Sigma-Aldrich. Anti-Myc (9E10, cat. TA150121Z, 1:400 for IP) and anti-HA (CB051, cat. TA180128-1, 1:400 for IP) were from Origene. Poly(dA:dT) and flagellin were purchased from InvivoGen. ML323 was obtained from Selleck Chemicals. CHX was purchased from APExBIO Technology. Anti-NLRP3 (polyclonal,

cat. 19771-1-AP, 1:400 for IP) was from Proteintech. Anti-NLRP3 (Cryo-2, cat. AG-20B-0014, 1:1000 for WB) and anti-Caspase-1 p20 (Casper-1, cat. AG-20B-0042, 1:1000) were obtained from AdipoGen. USP46 (polyclonal, cat. A17863, 1:1000) and USP12 (polyclonal, cat. A17862, 1:1000) antibodies were purchased from ABclonal Technology. Anti-Ub (P4D1, cat. sc-8017, 1:200), anti-β-actin (ACTBD11B7, cat. sc-8432, 1:200), protein A/G agarose used for immunoprecipitation and horseradish peroxidase-conjugated secondary antibodies were from Santa Cruz Biotechnology. Anti-Caspase-1 (14F468, cat. GTX14367, 1:1000) was from GenTex. Anti-p65 (D14E12, cat. #8242, 1:1000 for WB, 1:400 for IP), anti-p-p65 (Ser536, cat. #3033, 1:1000), anti-AIM2 (polyclonal, cat. #13095, 1:1000), anti-IL-1β/p17 (3A6, cat. #12242, 1:1000) and anti-USP1 (D37B4, cat. #8033, 1:1000 for WB, 1:400 for IP) were purchased from Cell Signaling Technology. Anti-UAF1 (polyclonal, cat. Ab122473, 1:1000 for WB, 1:400 for IP) and anti-NLRC4 (polyclonal, cat. Ab189593, 1:1000) were purchased from Abcam. The concentrations of agonists or stimulants were used as follows: LPS 200 ng/ml, ATP 5 mM, Nig 50 μM, poly (I:C) 10 μg/ml and flagellin 200 ng/ml. Poly(dA:dT) was transfected into macrophages with the final concentration as 200 ng/ml. ML323 30 mM or increasing amounts (15, 30, and 60 μM) were used in the cell experiments. In all the ubiquitination assays, MG132 (10 μM) was pretreated for 4 h before harvesting the cells.

**Cell culture**. To obtain mouse primary peritoneal macrophages, C57BL/6J mice (4–6 weeks old) were injected i.p. with 3% Brewer's thioglycollate broth. Three days later, peritoneal exudate cells (PECs) were harvested and incubated. Two hours later, nonadherent cells were discarded and adherent monolayer cells were used as peritoneal macrophages. Human embryonic kidney (HEK293T) cells were obtained from the American Type Culture Collection (Manassas, VA). Mouse embryonic fibroblasts (MEFs) were generated from female pregnant for 13–14 days mice[43]. In short, pregnant females were sacrificed by cervical dislocation, embryos were taken out, and the fetal viscera, head and limbs were excised. The rest of embryonic tissue was minced and incubated with 0.25% trypsin-EDTA for 30 min at 37 °C. MEFs were cultured and expanded for subsequent experiments. All the cells were cultured at 37 °C under 5% $CO_2$ in DMEM supplemented with 10% FCS (Invitrogen-Gibco), and 100 U/ml penicillin and 100 μg/ml streptomycin.

**Plasmids and transfection**. Expression plasmids for Flag-UAF1, Flag-USP1, Flag-USP12, TRAF6, TAK1, and TAB1 were constructed by PCR-based amplification of cDNA from THP-1 cells and cloned into the CMV-2 eukaryotic vector (Sigma-Aldrich)[36,44]. TRIF plasmids were gifts from X. Cao (Second Military Medical University, Shanghai, China). HA-Ub WT, HA-K48 Ub and HA-K63 Ub were gifted from H. Xiao (Institut Pasteur of Shanghai, Chinese Academy of Science, China). Myc-NLRP3 and Myc-ASC plasmids were gifts from Dr. John C. Reed (Sanford-Burnham Medical Research Institute, La Jolla, CA). The USP46 expression plasmid was procured from Sino biologicals, P65 expression plasmid from Vigene Biosciences and NF-κB reporter plasmid was purchased from Stratagene. NLRP3 mutants were generated using the KOD-Plus-Mutagenesis kit (Toyobo, Osaka, Japan, cat. SMK-101).

**RNA interference assay**. siRNAs were synthesized as following: 5'-GGUCGA-GACUCCAU CAUAA-3' (siRNA1) and 5'-GGAACAAAGACUCCAUUUA-3' (siRNA2) for *Uaf1*; 5'-GGCAAGUUAUGAGCUUAUA-3' (siRNA1) and 5'-CGGCAAGGUUGAAGAACAA-3' (siRNA 2) for *Usp1*; 5'-CCUAAUGACAGU-CUCCAAA-3' (siRNA1), 5'-CCCAAGAAGUUCAUCACAA-3' (siRNA2), and 5'-GCUUAAGAGGGUUCAGUAA-3' (siRNA3) for *Usp12*; 5'-GGGAACACUCA-CUAACGAA-3' (siRNA1), 5'-GCAUUA CAUCACCAUCGUA-3' (siRNA2), and 5'-GCUCAAGCCAUUGAGGAAU-3' (siRNA3) for *Usp46*; "scrambled" control sequences were 5'-UUCUCCGAACGUGUCACGU-3'. These siRNA duplexes were transfected into mouse peritoneal macrophages using INTERFERin reagents (PolyPlus, cat. 409-10) according to the manufacturer's instructions.

**ELISA**. The concentrations of mouse IL-1β (cat. 1210123), mouse TNF (cat. 1217203) and mouse IL-6 (cat. 1210602) were measured using ELISA kits (Dakewe Biotech Company Ltd., Shenzhen, China) according to the manufacturer's instructions.

**RT-PCR**. Total RNA was extracted using the RNA fast 200 RNA Extraction kit (Fastagen Biotech, Shanghai, China, cat. 220011) according to the manufacturer's instructions. RNA (500 ng) was reverse-transcribed using reverse transcriptase (Takara, cat. RR047A). A Light Cycler (Roche) and SYBR RT-PCR kit (Roche, cat. 06924204001) were used for RT-PCR analysis. Data were normalized to *Actb* expression in each sample. Specific primers used for real-time PCR assays are listed in Supplementary Table 1.

**Immunoprecipitation and immunoblot analysis**. For immunoprecipitation (IP), whole-cell extracts were lysed in IP buffer containing 1.0% (v/v) Nonidet P 40, 50 mM Tris-HCl pH 7.4, 50 mM EDTA, 150 mM NaCl, and a protease inhibitor 'cocktail' (Merck, cat. p8340). After centrifugation for 10 min at 14,000 *g*, supernatants were collected and incubated with protein G Plus-Agarose

Immunoprecipitation reagent together with a specific antibody. After 6 h of incubation, the beads were washed five times with IP buffer at 1000 × *g* for 5 min every time. Immunoprecipitates were eluted by boiling with 1% (w/v) SDS sample buffer. For re-IP, after the first IP, the beads were denatured by boiling in IP buffer containing 1% SDS. The eluates were diluted with IP buffer, and IP was performed similar to the first IP[43]. For immunoblot analysis, cells were lysed with M-PER Protein Extraction Reagent (Pierce, Rockford, IL, cat. 89901) supplemented with a protease inhibitor 'cocktail', and protein concentrations in the extracts were measured using a bicinchoninic acid assay (Pierce, Rockford, IL, cat. 23225). Equal amounts of extracts were separated by SDS-PAGE, and then transferred onto polyvinylidene-fluoride membranes for immunoblot analysis. Cell culture supernatants were harvested and concentrated for immunoblotting with Amicon Ultra 10 K (cat. UFC5010) from Millipore.

**In vivo LPS challenge**. C57BL/B6 mice were injected intraperitoneally with ML323 (10 mg/kg) or PBS. After 4 h, the mice were intraperitoneally injected with 10 mg/kg LPS or PBS. After 90 min, the mice were sacrificed and blood was collected for measurement of serum cytokines IL-1β, TNF, and IL-6 by ELISA. WT or $Uaf1^{CKO}$ mice (females, 6 weeks old) were intraperitoneally injected with 10 mg/kg LPS or PBS. After 2 h, the mice were sacrificed and blood was collected for measurement of serum cytokines IL-1β, TNF, and IL-6 by ELISA.

**In vivo FA-induced ATN**. C57BL/B6 mice were injected intraperitoneally with ML323 (10 mg/kg) or PBS. After 4 h, the mice were intraperitoneally injected with 250 mg/kg folic acid. After 12 h, the mice were sacrificed and blood was collected for estimation of serum cytokines IL-1β, TNF and IL-6 by ELISA. Furthermore, C57BL/B6 mice were injected intraperitoneally with ML323 (10 mg/kg) or PBS. After 4 h, the mice were intraperitoneally injected with 250 mg/kg folic acid. After 36 h, the mice were sacrificed, and the kidneys from the mice were dissected, partially fixed in 4% paraformaldehyde, embedded in paraffin, sectioned, stained with hematoxylin and eosin (H&E) and periodic-acid-Schiff (PAS), and examined by light microscopy for histologic changes. Renal inflammation was scored (based upon proportion of renal parenchyma by PAS staining) as not obvious (score 0), less than 10% (score 1), 10–25% (score 2), 25–50% (score 3), or more than 50% (score 4). Partial tissues were ground and disrupted for immunoblot analysis.

**NLRP3 inflammasome activation and analysis**. Mouse primary peritoneal macrophages were primed with LPS (Sigma-Aldrich, 200 ng/ml), followed by ATP (Sigma-Aldrich, 5 mM) or nigericin (Sigma-Aldrich, 50 μM) for 40 min. To evaluate NLRP3 inflammasome activation, IL-1β in the supernatant was detected by ELISA. The supernatants were concentrated using Amicon Ultra 10 K from Millipore to detect the expression of caspase-1 p20 and IL-1β p17.

**Immunofluorescence staining and confocal analysis**. MEFs transiently transfected with plasmids encoding Myc-NLRP3, GFP-USP1, or Flag-UAF1 were cultured for 24 h, and then stimulated with LPS for 2 h. NLRP3 and UAF1 were stained with a secondary antibody conjugated to either Alexa Fluor 633 or Alexa Fluor 488 (Molecular Probes, Invitrogen, polyclonal, cat. A-21070, A-11029) and nuclei were stained with DAPI (Molecular Probes, Invitrogen). The cells were examined by confocal laser microscopy (LSM780, Carl Zeiss).

**Assay of luciferase activity**. Luciferase activity was measured using the Dual-Luciferase Reporter Assay system according to the manufacturer's instructions (Promega)[36,43]. In short, HEK293T cells were cotransfected with the mixture of the indicated luciferase reporter plasmid, pRL-TK-Renilla-luciferase plasmid, and the indicated expression plasmids using Jet-PEI transfection reagent (Polyplus, cat. 101-40N). After 24 h, luciferase activities were measured with a Dual Luciferase Reporter Assay System (Promega, cat. E1960). Data were normalized for transfection efficiency by dividing firefly luciferase activity with that of Renilla luciferase.

**Statistical analysis**. All data are presented as mean ± SEM of three independent experiments. Statistical significance was determined using the two-tailed Student's *t* test, with a *p*-value < 0.05 considered statistically significant. All statistical analyses employed were performed using Prism 6.

**Reporting summary**. Further information on research design is available in the Nature Research Reporting Summary linked to this article.

## Data availability
The authors declare that the data supporting the findings of this study are available within the paper and its supplementary information files or available from the corresponding author upon reasonable request. Source data are provided with this paper.

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

## Acknowledgements

We thank Translational Medicine Core Facility of Shandong University for consultation and instrument availability that supported this work. This work was supported by grants from the National Natural Science Foundation of China (31570867, 81622030, 31870866, and 81861130369), and National Key Research and Developmental Program of China (2017YFC1001100). W.Z. is a Newton Advanced Fellow awarded by the Academy of Medical Sciences (NAF\R1\180232).

## Author contributions

W.Z. designed and supervised the research; H.S., C.Z., Z.Y., Q.L., R.Y., Y.Q., and M.J. performed the experiments; H.S., C.Z., and W.Z. analyzed the data and wrote the paper.

## Competing interests

The authors declare no competing interests.
