## [Peer Review File · Nature Communications]

Reviewers' comments:

Reviewer #1 (Remarks to the Author):

In this paper, Song et al found that the UAF1/USP1 deubiquitinase complex removes K-48-linked polyubiquitination of NLRP3 and inhibits NLRP3 degradation, resulting in the enhancement of NLRP3 inflammasome activity. They showed that (1) UAF1 deficiency and knockdown inhibited NLRP3 protein expression and NLRP3 inflammasome-driven IL-1b secretion; (2) UAF1/USP1 directly interacts with the LRR and NACHT domains of NLRP3; (3) UAF1/USP1 selectively removes K-48-linked NLRP3 ubiquitination and stabilizes NLRP3; (4) UAF1/USP12 and UAF1/USP46 promote NLRP3 expression at the transcription levels by targeting NF- κ B; (5) ML323 (a specific inhibitor for UAF1/USP1) inhibits IL-1b secretion and ameliorates the severity of FA-induced ATN in mice. In general, the experiments were well-executed and the data are clearly presented. The findings are novel and significant. However, the following concerns and questions that should be addressed to further strengthen the study.

Major comments:

1. The authors showed that UAF1/USP1 binds the LRR and NACHT domains of NLRP3 and deubiquitinates NLRP3. Can UAF1/USP1 deubiquitinate and regulate other PRRs (e.g., NLRP1, NLRC4, and AIM2) that form inflammasome assembly?
2. The authors showed that UAF1 directly interacts with USP1. Does UAF1 interact with USP12 and/or USP46? Do UAF1/USP12 and UAF1/USP46 remove K48- or K63-linked ubiquitination?
3. ML323 was used as a specific inhibitor for the UAF1/USP1 complex. What is the mechanism of ML323 action? Does ML323 inhibit association between UAF1 and USP1? Is the UAF1 and USP1 expression altered by ML323?
4. Fig. 1. The authors showed the cleavage of pro-caspase-1. The cleavage of IL-1b should be examined by western blotting.
5. The effect of LPS treatment on the expression levels of UAF1 should be examined.
6. In vivo experiments: The authors should evaluate a FA-induced ATN model in UAF1-CKO mice.
7. Figure 7: (c) The mRNA expression of IL-1b, IL-6, and TNF- α should be assessed. TNF- α protein secretion should also be assessed. (d) The protein levels and cleavage of IL-1b in the kidney tissue should be assessed. (f) The authors should perform PAS staining to assess tubular injury, and quantify the injury levels.

Minor comments:

Page 34: MG-132 (a proteasome inhibitor) was mentioned. However, there is no data regarding MG-132 in this study.

Reviewer #2 (Remarks to the Author):

The manuscript submitted by Song and colleagues describes a novel regulation of Nlrp3 inflammasome signaling by UAF1/USP deubiquitination complexes. The authors provide data arguing on the one hand for UAF1/USP12 and UAF1/USP46 complexes stabilizing p65 protein levels and thereby facilitating the Nlrp3 inflammasome priming step, and on the other hand arguing for the

UAF1/USP1 complex directly stabilizing the Nlrp3 protein itself. The study demonstrates that ML323, an inhibitor of the UAF1/USP1 complex, downregulates Nlrp3 inflammasome activity, which may be of broad interest to immunologists.

The findings presented in this manuscript are novel and interesting, but several claims are not sufficiently supported by the data provided and therefore require major revision before the manuscript can be accepted for publication. In particular, the authors focus their manuscript on Nlrp3 inflammasome activation but given the impact on p65 expression the effects of these UAF1/USP deubiquitination complexes are expected to be much broader, which the authors largely ignore (e.g. by omitting LPS controls in LPS/ATP inflammasome activation experiments and by not showing pro-IL-1b levels in these experiments). In addition, the data provided in support of the UAF1/USP deubiquitination mechanisms regulating p65 protein levels are not convincing, and in general many of the ubiquitination and protein-protein interaction experiments lack essential controls. Below are some specific suggestions on how to improve this manuscript.

Major remarks:

- The authors claim effects of UAF1/USP complexes on NF- κ B activation, but when analyzing Nlrp3 inflammasome activation they do not show the data in a manner that discriminates NF- κ B effects from Nlrp3 effects. In particular, since the authors use long priming of 8 hrs, LPS alone controls when measuring IL-1b levels should be shown (e.g. Fig 1a, Fig3a). In addition, in all western blotting analyses of Nlrp3 activation (Fig 2b, 2c, 3b, 3c, 3d) the levels of pro-IL-1b and active IL-1b should be shown to be able to reconcile whether the released IL-1b derives from enhanced pro-IL-1b production and/or pro-IL-1b processing. Moreover, these western blots should also contain Nlrp3 protein levels. In addition, in all LPS/ATP and LPS/Nig experiments secretion of IL-6 and TNF should be shown to demonstrate the (a)specificity of the UAF1/USP complex effects on Nlrp3 activation. The authors have provided some of these data in separate 'LPS only' experiments like in Fig 2 and Fig S2, but these analyses should be performed at once in the Nlrp3 inflammasome experiments to allow the reader to comprehensively appreciate the effects of the UAF1/USP complexes on inflammasome priming versus activation.
- Throughout the manuscript the authors analyse Nlrp3 inflammasome activation by blotting for caspase-1 p20 in the supernatant of the cells (e.g. Fig 2b, 2c, 3b, 3c, 3d, 7b). This implies that the authors are looking at the end result of lytic cell death, but the study does not contain any data on cell death. It is possible that disabling the function of the UAF1/USP complexes abrogates cell death without affecting caspase-1 cleavage, which would also result in less p20 in the supernatant. The authors need to provide analyses of lytic cell death in all experiments to allow the reader to discriminate cell death versus inflammasome activation.
- Instead of Fig 2f that uses physiologically irrelevant overexpression systems, the authors could use more NF- κ B activators in addition to LPS (other TLR agonists, TNF) in UAF1 ko cells to strengthen the claim that UAF1 has a uniform role in regulating NF- κ B activation.
- Throughout the manuscript, the authors should provide knockdown efficiency blots for each USP (e.g. Fig 3b-d). Although the authors provide such data in Sup Fig 1 and 3, knockdown efficiencies can differ between experiments and thus should be shown in each individual experiment.
- In general, the data in Fig 4 arguing that UAF1/USP12/46 complexes regulate p65 protein levels are not convincing. For instance, in Fig 4 c-e it was surprising to see that the effect on p65 stability in UAF1KO cells was milder when compared to the effect of USP12 or USP46 knockdown, since UAF1 is essential activating co-factor for both of these USPs. How do the authors explain this? In addition, in

Fig 4i the p65 input blot does not support the claim made in Fig 4e regarding the levels of p65 in UAF1 KO cell line. In this respect, it would also be helpful to see a total p65 blot in Fig 2d. Moreover, in Fig 4f the p65 blot on the IP samples shows surprisingly low amount of protein. Shouldn't one expect stronger p65 bands in the IP when compared to the input? Also in Fig 4f the input levels of USP46 increase over the LPS treatment, while its binding to p65 decreases. On the other hand, binding of USP12 to p65 increases with LPS. How do the authors reconcile this in their model? In this respect, it might also be good to include in the model the fact that activated p65 will shuttle from the cytoplasm to the nucleus. The authors used whole-cell extracts for all their IP's. Where are the USPs located and how do they travel between cytoplasm and nucleus upon LPS stimulation? Lastly, in Fig 4g the authors claim that USP1 does not bind to p65, but overexpression of USP1 is not sufficient enough in their input (when compared to USP12/46) to support this claim.

- In Fig 5. USP1 and UAF1 are never studied as a complex. Please provide evidence in Fig 5g and 5h that Nlrp3, USP1 and UAF1 are pulled down together in one endogenous complex. Aligning with this notion, the data in Fig 6i-j would be more convincing when ML323 would be combined with a proteasome inhibitor to show that loss of Nlrp3 is proteasome-mediated, and when a similar experiment would be performed in UAF1ko cells, as the ML323 effect would be predicted not to occur in UAF1ko conditions. The latter would greatly support the existence of a USP1/UAF1/Nlrp3 complex.

- Cohn et al (2007, Molecular Cell Vol 28:5, 786-797) has shown that each of the components of the USP1/UAF1 complex alone does not possess deubiquitination activity. Furthermore, UAF1 has no catalytic activity. Yet, the authors show in Fig 6a and 6b that overexpression of only one component is sufficient to remove Ub chains from Nlrp3. How do the authors explain this? In addition, how do the authors explain the discrepancies in the deubiquitination efficiencies of USP1 in Fig 6a vs UAF1 in Fig 6b; and of the drastic reduction in Ub by UAF1 in Fig 6b versus almost no effect in Fig 6f? Also in Fig 6f, one would expect that overexpression of K48 in the absence of UAF1 should destabilise the Nlrp3 protein, and hence Nlrp3 levels in the input blot would be expected to be less. Why is this not the case? In vitro deubiquitination assays could be a more convincing way to address the Ub linkage specificity of UAF1.

- The data provided would gain a lot of confidence by showing the effects of UAF1 on Nlrp3 and p65 levels also in vivo. The authors show functional differences in LPS and folic acid experiments in the UAF1cko mice, but they do not relate this to Nlrp3 and/or p65 levels in vivo. Can the authors assess Nlrp3 and p65 protein expression levels in myeloid cells from UAF1cko mice?

Minor comments:

- UAF1 knockdown data in Fig 1d-e and 2b and 2e can be removed since UAF1 knockout data are already provided.

- Molecular weight markers must be indicated in all ubiquitin blots. E.g. it is not possible to understand where the ubiquitin smears on p65 are in Fig 4h.

- Fig 4g. Mislabeling of IP blot, Myc instead of Flag?

- In the result section of Fig 7, b-g is wrongly referred.

- Sup Fig2b Nlrp3 mRNA graph is exactly the same as Fig 2c. Data should not be duplicated.

- Figs 3h and 4a would be more complete when also UAF1 siRNA would be included, and when an LPS+ML323 condition would be included.

- Are all control samples in ML323 experiments treated with DMSO? E.g. in Fig 3f such a DMSO control seems to be lacking.
- Fig 4i: Please rerun/reblot IP samples. There seems to be a shadow on the membrane that may affect the interpretation of the ubiquitination result.
- Fig 5 c-d. Include no Flag, with Myc control to exclude aspecific binding.
- Fig 6 c-d. Include Nlrp3 blot in IP samples to show IP efficiency.
- Fig 6e. Provide better blot for input Flag.
- In mouse experiments using the UAF1cko mice it is not mentioned what the control wild-type mice are exactly.
- Can the folic acid experiment of Fig 7g be performed in UAF1cko mice? Or does the folic acid act on non-myeloid cells?
- There are numerous grammatical mistakes in the manuscript. Please have your text edited by a native English speaker.

Reviewer #1

In this paper, Song et al found that the UAF1/USP1 deubiquitinase complex removes K-48-linked polyubiquitination of NLRP3 and inhibits NLRP3 degradation, resulting in the enhancement of NLRP3 inflammasome activity. They showed that (1) UAF1 deficiency and knockdown inhibited NLRP3 protein expression and NLRP3 inflammasome-driven IL-1b secretion; (2) UAF1/USP1 directly interacts with the LRR and NACHT domains of NLRP3; (3) UAF1/USP1 selectively removes K-48-linked NLRP3 ubiquitination and stabilizes NLRP3; (4) UAF1/USP12 and UAF1/USP46 promote NLRP3 expression at the transcription levels by targeting NF- κ B; (5) ML323 (a specific inhibitor for UAF1/USP1) inhibits IL-1b secretion and ameliorates the severity of FA-induced ATN in mice. In general, the experiments were well-executed and the data are clearly presented. The findings are novel and significant. However, the following concerns and questions that should be addressed to further strengthen the study.

Answer: We appreciate very much for your time in reviewing our manuscript. In accordance with your valuable suggestions and comments, we carefully revised the manuscript and several additional experiments were performed. In doing so, we have strengthened mechanistic details and the physiological relevance of our findings. The point-by-point answers to the comments and suggestions were listed as below.

Major comments:

1. The authors showed that UAF1/USP1 binds the LRR and NACHT domains of NLRP3 and deubiquitinates NLRP3. Can UAF1/USP1 deubiquitinate and regulate other PRRs (e.g., NLRP1, NLRC4, and AIM2) that form inflammasome assembly?

Answer: We accepted the valuable suggestions and examined the effects of UAF/USP1 on other PRRs. ML323 treatment, UAF1 deficiency and USP1 knockdown had no effect on the expression of AIM2 and NLRC4, another two PRRs that could form inflammasome. We added these new data in the revised manuscript.

2. The authors showed that UAF1 directly interacts with USP1. Does UAF1 interact with USP12 and/or USP46? Do UAF1/USP12 and UAF1/USP46 remove K48- or K63-linked ubiquitination?

Answer: In addition to USP1, UAF1 also binds to USP12 and USP46. UAF1 constitutes three deubiquitinating enzyme complexes, including UAF1/USP1, UAF1/USP12, and UAF1/USP46. UAF1 constitutively binds to USP1, USP12 and USP46, and this binding greatly enhances their deubiquitinase activity (Cohn et al., *J Biol Chem.* 2009.; Yin et al., *Structure.* 2015.; Dharadhar et al., *J Struct Biol.* 2016.). It has been reported that USP46 removes K63-Ub of AMPA receptor (Huo et al., *J Neurochem.* 2015.). We found that USP12/46 selectively removed K48-Ub of p65, with no effect on K63-Ub of p65. Therefore, these results indicate that the UAF1/USP12 and UAF1/USP46 complexes remove K48-linked ubiquitination of p65. We added these new data in the revised manuscript.

3. ML323 was used as a specific inhibitor for the UAF1/USP1 complex. What is the mechanism of ML323 action? Does ML323 inhibit association between UAF1 and USP1? Is the UAF1 and USP1 expression altered by ML323?

Answer: ML323 is an allosteric inhibitor of USP1-UAF1, and dose not disrupt the USP1-UAF1 complex (Liang et al. *Nat Chem Biol.* 2014.). We added the information in the revised manuscript.

The majority of USP1 is complexed with UAF1 *in vivo* and the interaction of UAF1 with USP1 is necessary for the stability of USP1 *in vivo* (Cohn et al., *Mol Cell.* 2007.). UAF1 constitutively binds to USP1, and this binding greatly enhances USP1 deubiquitinase activity. Thus, the enzymatic activity of the USP1-UAF1 complex may be important for the stability of these two proteins.

4. Fig. 1. The authors showed the cleavage of pro-caspase-1. The cleavage of IL-1b should be examined by western blotting.

Answer: We accepted the valuable suggestions and examined the cleavage of pro-caspase-1/IL-1 β by Western blot. We added these new data in the revised manuscript.

5. The effect of LPS treatment on the expression levels of UAF1 should be examined.

Answer: Previously, we reported that LPS stimulation induced UAF1 expression in macrophages (Yu et al., *J Exp Med.* 2017). Thus, UAF1 is a feedback enhancer of NLRP3 inflammasome activation. We cited it and discussed this issue in the revised manuscript.

6. In vivo experiments: The authors should evaluate a FA-induced ATN model in UAF1-CKO mice.

Answer: We accepted the useful suggestions and evaluated FA-induced ATN using *Uaf1*-CKO mice. *Uaf1* deficiency in myeloid cells markedly suppressed IL-1 β secretion in kidneys (Fig. 7i). Less severe renal inflammation and edema was observed in *Uaf1*^{CKO} mice (Fig. 7j and 7k). These data indicate that *Uaf1* deficiency in myeloid cells ameliorates FA-induced ATN and suggest UAF1 as a physiological enhancer of inflammation. We added these new data in the revised manuscript.

7. Figure 7: (c) The mRNA expression of IL-1b, IL-6, and TNF-α should be assessed. TNF-α protein secretion should also be assessed.

Answer: We appreciated for the useful suggestions. We examined TNF-α secretion in serum of LPS-stimulated mice. ML323 treated mice produced less TNF-α and IL-1β in sera after i.p. injection of LPS than control mice. These data indicate that ML323 is active *in vivo*. In addition, the mRNA level of these cytokines are hard to detect in blood. Therefore, we did not assess the mRNA expression of these cytokines.

(d) The protein levels and cleavage of IL-1b in the kidney tissue should be assessed. (f) The authors should perform PAS staining to assess tubular injury, and quantify the injury levels.

Answer: We examined the cleavages of IL-1β in kidneys. *Uaf1* deficiency in myeloid cells markedly suppressed IL-1β secretion in kidneys (Fig.7i). In addition, we performed PAS staining and quantified the levels of renal inflammation. Renal inflammation was greatly ameliorated in ML323 treated or *Uaf1^{CKO}* mice (Fig. 7g and 7k). We added these new data in the revised manuscript.

Minor comments:

Page 34: MG-132 (a proteasome inhibitor) was mentioned. However, there is no data regarding MG-132 in this study.

Answer: We are sorry for the missing of MG-132 information used in several experiments. Because USP-UAF1 complex enhance the expression of its targets by inhibiting protein degradation. To better evaluate the effects of USP-UAF1 complex on the ubiquitination of its targets (including NLRP3 and p65), we pretreated with MG-132 for 4 h before harvesting cells in all the ubiquitination experiments. We added these information in the Methods section in the revised manuscript.

Reviewer #2

The manuscript submitted by Song and colleagues describes a novel regulation of Nlrp3 inflammasome signaling by UAF1/USP deubiquitination complexes. The authors provide data arguing on the one hand for UAF1/USP12 and UAF1/USP46 complexes stabilizing p65 protein levels and thereby facilitating the Nlrp3 inflammasome priming step, and on the other hand arguing for the UAF1/USP1 complex directly stabilizing the Nlrp3 protein itself. The study demonstrates that ML323, an inhibitor of the UAF1/USP1 complex, downregulates Nlrp3 inflammasome activity, which may be of broad interest to immunologists.

The findings presented in this manuscript are novel and interesting, but several claims are not sufficiently supported by the data provided and therefore require major revision before the manuscript can be accepted for publication. In particular, the authors focus their manuscript on Nlrp3 inflammasome activation but given the impact on p65 expression the effects of these UAF1/USP deubiquitination complexes are expected to be much broader, which the authors largely ignore (e.g. by omitting LPS controls in LPS/ATP inflammasome activation experiments and by not showing pro-IL-1 β levels in these experiments). In addition, the data provided in support of the UAF1/USP deubiquitination mechanisms regulating p65 protein levels are not convincing, and in general many of the ubiquitination and protein-protein interaction experiments lack essential controls. Below are some specific suggestions on how to improve this manuscript.

Answer: We appreciate very much for your time in reviewing our manuscript. In accordance with your valuable suggestions and comments, we carefully revised the manuscript and several additional experiments were performed. In doing so, we have strengthened mechanistic details and the physiological relevance of our findings. The point-by-point answers to the comments and suggestions were listed as below.

Major remarks:

- The authors claim effects of UAF1/USP complexes on NF- κ B activation, but when analyzing Nlrp3 inflammasome activation they do not show the data in a manner that discriminates NF- κ B effects from Nlrp3 effects.

In particular, since the authors use long priming of 8 hrs, LPS alone controls when measuring IL-1 β levels should be shown (e.g. Fig 1a, Fig3a).

Answer: We accepted the valuable suggestions and measured IL-1 β secretion in LPS stimulated WT and UAF1 deficient macrophages. IL-1 β level is too low to detect in only LPS treated macrophages. Therefore, we included these data in Fig.1a and did not examine IL-1 β secretion in Fig.3a.

In addition, in all western blotting analyses of Nlrp3 activation (Fig 2b, 2c, 3b, 3c, 3d) the levels of pro-IL-1b and active IL-1b should be shown to be able to reconcile whether the released IL-1b derives from enhanced pro-IL-1b production and/or pro-IL-1b processing. Moreover, these western blots should also contain Nlrp3 protein levels.

Answer: We appreciated the valuable suggestions and examined the protein levels of pro-IL-1 β , mature IL-1 β and NLRP3. We added these results in the revised manuscript.

In addition, in all LPS/ATP and LPS/Nig experiments secretion of IL-6 and TNF should be shown to demonstrate the (a)specificity of the UAF1/USP complex effects on Nlrp3 activation. The authors have provided some of these data in separate 'LPS only' experiments like in Fig 2 and Fig S2, but these analyses should be performed at once in the Nlrp3 inflammasome experiments to allow the reader to comprehensively appreciate the effects of the UAF1/USP complexes on inflammasome priming versus activation.

Answer: We appreciated for the valuable suggestion and performed additional experiments. UAF1 deficiency significantly attenuated LPS, LPS plus ATP, and LPS plus Nig induced IL-6 and TNF- α secretion (Fig.1). We added these new data in the revised manuscript. In addition, UAF1 enhanced LPS-induced NLRP3, IL-6 and TNF- α mRNA expression (Fig.2b and Supplementary Fig.3), and NF- κ B activation (Fig.2d-e). These data indicate that UAF1 enhanced the activation of both NLRP3 inflammasome and TLR4.

However, USP1 knockdown had no effects on LPS-induced NLRP3, IL-1 β , IL-6 and TNF- α mRNA expression (Fig.3h and 4a), and NF- κ B activation (Fig.4b and Supplementary Fig.5a). These data indicate that USP1 had no effect on the priming step of NLRP3 inflammasome activation. But, USP12 or USP46 enhanced LPS-induced NLRP3, IL-1 β , IL-6 and TNF- α mRNA expression (Fig.3h and 4a), and NF- κ B activation (Fig.4b and Supplementary Fig.5a). These data indicate that USP12/USP46 enhance the priming step of NLRP3 inflammasome activation.

Taken together, these data demonstrate that USP12/46 promotes the priming step of NLRP3 inflammasome activation, whereas USP1 enhances the activation step. Thus, UAF1 enhances both the priming and activation step of NLRP3 inflammasome via USP1/12/46.

- Throughout the manuscript the authors analyse Nlrp3 inflammasome activation by blotting for caspase-1 p20 in the supernatant of the cells (e.g. Fig 2b, 2c, 3b, 3c, 3d, 7b). This implies that the authors are looking at the end result of lytic cell death, but the study does not contain any data on cell death. It is possible that disabling the function of the UAF1/USP complexes abrogates cell death without affecting caspase-1 cleavage, which would also result in less p20 in the supernatant. The authors need to provide analyses of lytic cell death in all experiments to allow the reader to discriminate cell death versus inflammasome activation.

Answer: Yes, this a great question. We performed CCK8 assay to examine whether UAF1 deficiency or ML323 treatment could affect the cell viability. No differences of cell viability were observed in DMSO or ML323 pretreated macrophages, followed LPS stimulation or NLRP3 inflammasome activation. Similar results were observed in UAF1 deficient macrophages. These data indicate that UAF1 complex could not affect the lytic cell death induced by NLRP3 inflammasome activation.

- Instead of Fig 2f that uses physiologically irrelevant overexpression systems, the authors could use more NF- κ B activators in addition to LPS (other TLR agonists, TNF) in UAF1 ko cells to strengthen the claim that UAF1 has a uniform role in regulating NF- κ B activation.

Answer: Yes, this a great suggestion. We examined another TLR agonist induced NF- κ B activation in *Uaf1* KO macrophages and found that UAF1 deficiency inhibited poly(I:C)-induced p65 phosphorylation. We added these new data in the revised manuscript.

- Throughout the manuscript, the authors should provide knockdown efficiency blots for each USP (e.g. Fig 3b-d). Although the authors provide such data in Sup Fig 1 and 3, knockdown efficiencies can differ between experiments and thus should be shown in each individual experiment.

Answer: We appreciated the valuable suggestion and provided the knockdown efficiencies in each knockdown experiments in the revised manuscript.

- In general, the data in Fig 4 arguing that UAF1/USP12/46 complexes regulate p65 protein levels are not convincing. For instance, in Fig 4 c-e it was surprising to see that the effect on p65 stability in UAF1KO cells was milder when compared to the effect of USP12 or USP46 knockdown, since UAF1 is essential activating co-factor for both of these USPs. How do the authors explain this?

Answer: The phenomenon may cause by individual difference of mice. To better elucidate the effects of UAF1, we replaced Fig.4e, in which the experiment was performed using different mice.

In addition, in Fig 4i the p65 input blot does not support the claim made in Fig 4e regarding the levels of p65 in UAF1 KO cell line. In this respect, it would also be helpful to see a total p65 blot in Fig 2d.

Answer: Because USP-UAF1 complex enhance the expression of its targets by inhibiting protein degradation. To better evaluate the effects of USP-UAF1 complex on the ubiquitination of its targets, including NLRP3 and p65, we pretreated with MG132 for 4 h before harvesting cells in all the ubiquitination experiments. Thus, no difference of p65 level was observed in Input blot of ubiquitination related experiments. We are sorry for the missing of MG132 information used in several experiments and have added these information in the Methods section in the revised manuscript. In addition, we examined p65 expression in UAF1 deficient macrophages and found that UAF1 deficiency inhibited total p65 expression in macrophages (Fig.4e).

Moreover, in Fig 4f the p65 blot on the IP samples shows surprisingly low amount of protein. Shouldn't one expect stronger p65 bands in the IP when compared to the input? Also in Fig 4f the input levels of USP46 increase over the LPS treatment, while its binding to p65 decreases. On the other hand, binding of USP12 to p65 increases with LPS. How do the authors reconcile this in their model? In this respect, it might also be good to include in the model the fact that activated p65 will shuttle from the cytoplasm to the nucleus. The authors used whole-cell extracts for all their IP's. Where are the USPs located and how do they travel between cytoplasm and nucleus upon LPS stimulation?

Answer: This is very important comments and suggestions. The differences between IP and Input samples may cause by different amounts. We re-Blotted these samples and presented new figure in the revised manuscript. Previously, we found that USP1 immediately translocated from the nucleus to

cytoplasm after LPS stimulation in macrophages (Yu et al., *J Exp Med.* 2017). We can perform the suggested experiments about USP12/USP46 and p65 translocation in future studies.

Fig. 4f

Lastly, in Fig 4g the authors claim that USP1 does not bind to p65, but overexpression of USP1 is not sufficient enough in their input (when compared to USP12/46) to support this claim.

Answer: Thanks for the comment. To better address this conclusion, we re-Blotted the samples and presented a new blot for input Flag of Fig.4g in the revised manuscript.

Fig. 4g

- In Fig 5. USP1 and UAF1 are never studied as a complex. Please provide evidence in Fig 5g and 5h that Nlrp3, USP1 and UAF1 are pulled down together in one endogenous complex. Aligning with this notion, the data in Fig 6i-j would be more convincing when ML323 would be combined with a proteasome inhibitor to show that loss of Nlrp3 is proteasome-mediated, and when a similar experiment would be performed in UAF1ko cells, as the ML323 effect would be predicted not to occur in UAF1ko conditions. The latter would greatly support the existence of a USP1/UAF1/Nlrp3 complex.

Answer: This is a good suggestion. We performed the IP experiment as suggested. We found that USP1 could interact both UAF1 and NLRP3 (Fig.5g), suggesting that USP1 and UAF1 function as a complex. To better confirm it, we can perform the suggested the ‘MG132 plus’ experiments in *Uaf1* deficient macrophages. *Uaf1* deficiency lost the inhibitory effects on NLRP3 expression in MG132 treated macrophages (Fig.6k). We added these new data in the revised manuscript.

- Cohn et al (2007, Molecular Cell Vol 28:5, 786-797) has shown that each of the components of the USP1/UAF1 complex alone does not possess deubiquitination activity. Furthermore, UAF1 has no catalytic activity. Yet, the authors show in Fig 6a and 6b that overexpression of only one component is sufficient to remove Ub chains from Nlrp3. How do the authors explain this?

Answer: Yes, this a great question. In the overexpression experiments, we used HEK293T cells, which have endogenous USP1 and UAF1 expression. Thus, overexpression of one component of the USP1-UAF1 complex could enhance the catalytic activity of the complex.

In addition, how do the authors explain the discrepancies in the deubiquitination efficiencies of USP1 in Fig 6a vs UAF1 in Fig 6b; and of the drastic reduction in Ub by UAF1 in Fig 6b versus almost no effect in Fig 6f?

Answer: Yes, this is a good question and we also notice this phenomenon. The deubiquitination

efficiencies may cause by the HEK293T cells used in each experiment are different generation, which can affect the transfection efficiency and cell activity. However, we believed that the discrepancy would not affect the conclusion because the HEK293T cells used in every experiment are from same generation.

Also in Fig 6f, one would expect that overexpression of K48 in the absence of UAF1 should destabilise the Nlrp3 protein, and hence Nlrp3 levels in the input blot would be expected to be less. Why is this not the case?

Answer: Because USP-UAF1 complex enhance the expression of its targets by inhibiting proteasomal degradation. To better evaluate the effects of USP-UAF1 complex on the ubiquitination of its targets, including NLRP3 and p65, we pretreated with MG132 for 4 h before harvesting cells in all the ubiquitination experiments. Thus, no difference of NLRP3 level was observed in Input blot of ubiquitination related experiments. We are sorry for the missing of MG132 information used in several experiments and have added these information in the Methods section in the revised manuscript.

In vitro deubiquitination assays could be a more convincing way to address the Ub linkage specificity of UAF1.

Answer: This is a great suggestion. It would be more convincing to address the Ub linkage specific of UAF1 via in vitro assays. However, it would be more difficult to perform in vitro deubiquitination assays than in vitro ubiquitination assay, because the NLRP3 protein generated by in vitro translation possessed no-Ub linkage. Therefore, to address the specificity and exclude the effect of other proteins in complex, we performed a two-step immunoprecipitation assay (Re-IP). In HEK293T cells, Myc-tagged NLRP3 was cotransfected with ubiquitin and UAF1, respectively. The cell lysates were subjected to immunoprecipitation with anti-Myc, and then the immunoprecipitates were denatured, followed by reimmunoprecipitation with anti-Myc. NLRP3 polyubiquitination was markedly attenuated in the presence of UAF1 (Fig. 6d).

Fig. 6d

- The data provided would gain a lot of confidence by showing the effects of UAF1 on Nlrp3 and p65 levels also in vivo. The authors show functional differences in LPS and folic acid experiments in the UAF1cko mice, but they do not relate this to Nlrp3 and/or p65 levels in vivo. Can the authors assess Nlrp3 and p65 protein expression levels in myeloid cells from UAF1cko mice?

Answer: This is a great suggestion. We examined the cleavages of IL-1 β in kidneys. *Uaf1* deficiency in myeloid cells markedly suppressed IL-1 β secretion in kidneys (Fig.7i). Non-myeloid cells in kidney constitutively express NLRP3 and p65. Because *Uaf1* is deficient in myeloid cells, but not all kidney cells, we didn't examine the total NLRP3 and p65 expression in the model.

Fig. 7i

Minor comments:

- UAF1 knockdown data in Fig 1d-e and 2b and 2e can be removed since UAF1 knockout data are already provided.

Answer: We accepted the useful suggestion and removed the UAF1 knockdown data from main figures to Supplementary data.

- Molecular weight markers must be indicated in all ubiquitin blots. E.g. it is not possible to

understand where the ubiquitin smears on p65 are in Fig 4h.

Answer: We accepted the valuable suggestions and indicated the molecular weight marker of the Ub results in the revised manuscript.

- Fig 4g. Mislabeling of IP blot, Myc instead of Flag?

Answer: We are sorry for the mistake and corrected it in the revised manuscript.

- In the result section of Fig 7, b-g is wrongly referred.

Answer: We are sorry for the mistake and corrected it in the revised manuscript.

- Sup Fig2b Nlrp3 mRNA graph is exactly the same as Fig 2c. Data should not be duplicated.

Answer: We accepted the useful suggestion and deleted the NLRP3 mRNA graph in the Supplementary Figures.

- Figs 3h and 4a would be more complete when also UAF1 siRNA would be included, and when an LPS+ML323 condition would be included.

Answer: We have investigated the effects of UAF1 deficiency on the mRNA expression of NLRP3 (Fig.2b), TNF- α and IL-6 (Supplementary Fig.3). Thus, we have not examine the effects of UAF1 siRNA on the mRNA expression of NLRP3, TNF- α and IL-6. We performed additional experiments to examine the effects of UAF1 deficiency on the mRNA expression of IL-1 β , and found that UAF1 deficiency inhibited LPS-induced IL-1 β mRNA expression (Supplementary Fig.3).

- Are all control samples in ML323 experiments treated with DMSO? E.g. in Fig 3f such a DMSO control seems to be lacking.

Answer: In ML323 experiments, all control samples are treated with DMSO, including Fig.3f. We are

sorry for the missing of these information and added them in the revised manuscript.

- Fig 4i: Please rerun/reblot IP samples. There seems to be a shadow on the membrane that may affect the interpretation of the ubiquitination result.

Answer: We accepted the suggestion and represented it in the revised manuscript.

- Fig 5 c-d. Include no Flag, with Myc control to exclude aspecific binding.

Answer: We accepted the valuable suggestion and repeated the experiments with suggested control.

We provided these new data in the revised manuscript.

- Fig 6 c-d. Include Nlrp3 blot in IP samples to show IP efficiency.

Answer: We accepted the valuable suggestion and provided NLRP3 blot in IP samples in the revised manuscript.

- Fig 6e. Provide better blot for input Flag.

Answer: We accepted the useful suggestion and provided better blot for input Flag it in the revised manuscript.

- In mouse experiments using the UAF1cko mice it is not mentioned what the control wild-type mice are exactly.

Answer: For animal experiments with *Uaf1*^{CKO} mice, littermate controls with normal *Uaf1* expression (*Uaf1*^{fl^{ox}/fl^{ox}}) were used. We provided this information in the Methods section in the revised manuscript.

- Can the folic acid experiment of Fig 7g be performed in UAF1cko mice? Or does the folic acid act on non-myeloid cells?

Answer: We accepted the useful suggestions and evaluated FA-induced ATN using *Uaf1*-CKO mice.

Uaf1 deficiency in myeloid cells markedly suppressed IL-1 β secretion in kidneys (Fig. 7i). Less severe renal inflammation and edema were observed in *Uaf1*^{CKO} mice (Fig. 7j and 7k). These data indicate that *Uaf1* deficiency in myeloid cells ameliorates FA-induced ATN and suggest UAF1 as a physiological enhancer of inflammation. We added these new data in the revised manuscript.

- There are numerous grammatical mistakes in the manuscript. Please have your text edited by a native English speaker

Answer: We accepted the valuable suggestion and have performed language editing through Editage (www.editage.com).

REVIEWER COMMENTS

Reviewer #1 (Remarks to the Author):

The authors responded to the reviewer's comments adequately, and this reviewer has no further comments for this paper.

Reviewer #2 (Remarks to the Author):

I appreciate very much the great efforts made by the authors to address my comments. Overall, I believe the manuscript improved a lot and I think it would merit publication provided that the authors give sufficient credit to their new data showing that UAF1/Usp12 and UAF1/Usp46 in fact promote NF- κ B activation and thus prime the inflammasome by augmenting not only Nlrp3 but also pro-IL-1b levels, and moreover showing that the effects of UAF1 are not specific to inflammasome responses (given the new TNF and IL-6 data). My detailed comments on the new data are provided below.

1. The new experiments requested by me as well as by reviewer 1 to discriminate between UAF1 effects on inflammasome priming vs activation show decreased LPS-induced mRNA as well as protein expression levels of TNF, IL-6, pro-IL-1b and Nlrp3 in the absence of UAF1 (Fig 1A, 1B, 1C, 1D, S2C, S3A, S3B). These novel data very convincingly demonstrate that reduced UAF1 expression is associated with diminished NF- κ B-dependent gene expression, which results in decreased priming of inflammasome responses because of insufficient LPS-induced Nlrp3 and pro-IL-1b production. This is in line with the observations of the authors that UAF1/Usp12 and UAF1/Usp46 complexes regulate p65 levels. On the other hand, the authors show that the UAF1/Usp1 complex regulates Nlrp3 expression post-translationally by removing K48-linked Ub chains from Nlrp3. However, despite their new pro-IL-1b, IL-6 and TNF data, the authors largely ignore the pro-IL-1b priming effects of UAF1 as well as the inflammasome-independent effects and focus the abstract and the discussion of their manuscript on the effects of UAF1 on Nlrp3 expression. The effect of UAF1 on pro-IL-1b levels was in fact much more prominent and likely contributes more to the authors' observations than the effects on Nlrp3 expression levels. For instance, subtle siRNA knockdown of UAF1 clearly decreased pro-IL-1b levels but hardly affected Nlrp3 expression levels in Fig S2C. In addition, throughout the study the effect of UAF1 on pro-IL-1b cleavage is more convincing than its effect of caspase-1 cleavage.

I believe the authors should be a bit less biased and re-write the manuscript (especially the abstract and discussion) to acknowledge the differential contributions of NF- κ B activation effects and Nlrp3 protein stability effects in regulating Nlrp3 inflammasome outcomes. In addition, the authors should also clearly acknowledge the effects of UAF1 on other important cytokines such as IL-6 and TNF, especially in their in vivo experiments. Related to this I have two questions on newly provided data:

1a. In response to a question of reviewer 1, the authors show in Fig 2C that LPS-induced Nlrp4 and Aim2 expression was not altered in UAF1 deficiency, thus suggesting that the effects of UAF1 are specific for Nlrp3 activation. It is indeed known that Nlrp4 and Aim2 expression are not enhanced by LPS. However, given the effects of UAF1 on LPS-induced pro-IL-1b levels the Nlrp4 and Aim2 data are a bit misleading because in a normal physiological situation (e.g. Salmonella infection for Nlrp4 activation and Francisella infection for Aim2 activation) UAF1-deficient cells would likely induce less pro-IL-1b leading to diminished Nlrp4- and Aim2-dependent inflammasome outputs. Can the authors perform Nlrp4 and Aim2 inflammasome activation assays evaluating IL-1b secretion to determine the real specificity of UAF1 in regulating the cytokine outcomes of activating different inflammasomes?

1b. The authors provided a new folic acid in vivo experiment in Fig 7i-k nicely showing that myeloid

UAF1 deficiency prevents kidney injury. However, they do not discriminate whether these effects are due to decreasing Nlrp3 expression, decreasing pro-IL-1b expression and/or decreasing TNF or IL-6 levels. Instead, the authors only show cleaved IL-1b levels in a western blot in Fig 7i. Can the authors supplement this western blot with pro-IL-1b and Nlrp3 expression levels, and can the authors provide cytokine levels for IL-6 and TNF?

Minor comments:

- The authors could be a bit more careful in a couple of statements. E.g. on lines 84-85 'our study thus uncovers the mechanism of Nlrp3 inflammasome activation' should probably be '... some additional mechanisms regulating Nlrp3 ...'. And on lines 303-308 the authors claim ML323 as 'a novel Nlrp3 inhibitor' and 'a promising candidate for the treatment of inflammatory diseases caused by aberrant Nlrp3 activity'. ML323 does many things and this statement is not in agreement with the NF- κ B effects observed by the authors.
- The title of Fig Legend S5 should be 'USP1 has no effect on NF- κ B activation or p65 expression, and Usp12/46 have no effects on K63-linked Ub of p65'.

Reviewer #2 (Remarks to the Author):

I appreciate very much the great efforts made by the authors to address my comments. Overall, I believe the manuscript improved a lot and I think it would merit publication provided that the authors give sufficient credit to their new data showing that UAF1/Usp12 and UAF1/Usp46 in fact promote NF-kB activation and thus prime the inflammasome by augmenting not only Nlrp3 but also pro-IL-1b levels, and moreover showing that the effects of UAF1 are not specific to inflammasome responses (given the new TNF and IL-6 data). My detailed comments on the new data are provided below.

1. The new experiments requested by me as well as by reviewer 1 to discriminate between UAF1 effects on inflammasome priming vs activation show decreased LPS-induced mRNA as well as protein expression levels of TNF, IL-6, pro-IL-1b and Nlrp3 in the absence of UAF1 (Fig 1A, 1B, 1C, 1D, S2C, S3A, S3B). These novel data very convincingly demonstrate that reduced UAF1 expression is associated with diminished NF-kB-dependent gene expression, which results in decreased priming of inflammasome responses because of insufficient LPS-induced Nlrp3 and pro-IL-1b production. This is in line with the observations of the authors that UAF1/Usp12 and UAF1/Usp46 complexes regulate p65 levels. On the other hand, the authors show that the UAF1/Usp1 complex regulates Nlrp3 expression post-translationally by removing K48-linked Ub chains from Nlrp3. However, despite their new pro-IL-1b, IL-6 and TNF data, the authors largely ignore the pro-IL-1b priming effects of UAF1 as well as the inflammasome-independent effects and focus the abstract and the discussion of their manuscript on the effects of UAF1 on Nlrp3 expression. The effect of UAF1 on pro-IL-1b levels was in fact much more prominent and likely contributes more to the authors' observations than the effects on Nlrp3 expression levels. For instance, subtle siRNA knockdown of UAF1 clearly decreased pro-IL-1b levels but hardly affected Nlrp3 expression levels in Fig S2C. In addition, throughout the study the effect of UAF1 on pro-IL-1b cleavage is more convincing than its effect of caspase-1 cleavage.

I believe the authors should be a bit less biased and re-write the manuscript (especially the abstract and discussion) to acknowledge the differential contributions of NF-kB activation effects and Nlrp3 protein stability effects in regulating Nlrp3 inflammasome outcomes. In addition, the authors should also clearly acknowledge the effects of UAF1 on other important cytokines such as IL-6 and TNF, especially in their in vivo experiments.

Answer: Thanks for the valuable suggestions and the manuscript has been revised accordingly. We

added “the UAF1/USP12 and UAF1/USP46 complexes promote NF- κ B activation, enhance the transcription of NLRP3 and proinflammatory cytokines (including pro-IL-1 β , TNF- α and IL-6) by inhibiting ubiquitous degradation of p65” in the abstract and discussion section.

Related to this I have two questions on newly provided data:

1a. In response to a question of reviewer 1, the authors show in Fig 2C that LPS-induced Nlr4 and Aim2 expression was not altered in UAF1 deficiency, thus suggesting that the effects of UAF1 are specific for Nlr3 activation. It is indeed known that Nlr4 and Aim2 expression are not enhanced by LPS. However, given the effects of UAF1 on LPS-induced pro-IL-1b levels the Nlr4 and Aim2 data are a bit misleading because in a normal physiological situation (e.g. Salmonella infection for Nlr4 activation and Francisella infection for Aim2 activation) UAF1-deficient cells would likely induce less pro-IL-1b leading to diminished Nlr4- and Aim2-dependent inflammasome outputs. Can the authors perform Nlr4 and Aim2 inflammasome activation assays evaluating IL-1b secretion to determine the real specificity of UAF1 in regulating the cytokine outcomes of activating different inflammasomes?

Answer: This is a great suggestion and we performed the suggested experiment. *Uaf1* deficiency decreased AIM2 inflammasome and NLRC4 inflammasome activation induced IL-1 β secretion (Supplementary Fig. 5c). We added these new data in the revised manuscript.

Supplementary Fig. 5c

1b. The authors provided a new folic acid in vivo experiment in Fig 7i-k nicely showing that myeloid UAF1 deficiency prevents kidney injury. However, they do not discriminate whether these effects are due to decreasing Nlr3 expression, decreasing pro-IL-1b expression and/or decreasing TNF or IL-6 levels. Instead, the authors only show cleaved IL-1b levels in a western blot in Fig 7i. Can the authors supplement this western blot with pro-IL-1b and Nlr3 expression levels, and can the authors provide cytokine levels for IL-6 and TNF?

Answer: We accepted the valuable suggestion and examined pro-IL-1 β expression in kidney. *Uaf1* deficiency in myeloid cells also suppressed pro-IL-1 β expression in the kidneys (Fig.7i). We added these new data in the revised manuscript. However, *Uaf1* deficiency in myeloid cells did not inhibit NLRP3 expression in the kidneys (data not shown). This phenomenon may cause by the effect that the renal parenchymal cells also express NLRP3. Unfortunately, we did not examine the IL-6 and TNF levels in kidneys because they are hard to examine in kidney tissues.

Minor comments:

- The authors could be a bit more careful in a couple of statements. E.g. on lines 84-85 ‘our study thus uncovers the mechanism of Nlrp3 inflammasome activation’ should probably be ‘... some additional mechanisms regulating Nlrp3 ...’.

Answer: We accepted the great suggestion and revised the statements accordingly.

And on lines 303-308 the authors claim ML323 as ‘a novel Nlrp3 inhibitor’ and ‘a promising candidate for the treatment of inflammatory diseases caused by aberrant Nlrp3 activity’. ML323 does many things and this statement is not in agreement with the NF- κ B effects observed by the authors.

Answer: ML323 is a selective inhibitor of UAF1/USP1 complex, with no inhibitory effects on both UAF1/USP12 and UAF1/USP46 complexes (Liang, Q. et al. A selective USP1-UAF1 inhibitor links deubiquitination to DNA damage responses. *Nat Chem Biol* 2014; 10, 298-304.). Although UAF1 enhances NF- κ B activation, USP1 and ML323 both had no effects on NF- κ B activation. We added the information in the discussion section.

- The title of Fig Legend S5 should be ‘USP1 has no effect on NF- κ B activation or p65 expression, and Usp12/46 have no effects on K63-linked Ub of p65’.

Answer: We are sorry for the inaccurate statement. We corrected it as suggestion in the revised manuscript.

REVIEWER COMMENTS

Reviewer #2 (Remarks to the Author):

In my previous recommendations to the authors, I had asked to 'be a bit less biased and re-write the manuscript (especially the abstract and discussion) to acknowledge the differential contributions of NF-kB activation effects and Nlrp3 protein stability effects in regulating Nlrp3 inflammasome outcomes'. Instead of re-writing 'especially' the abstract and the discussion, the authors decided to re-write 'only' the abstract and the discussion (by adding 1 sentence to each). The manuscript continues to mislead the reader by focusing on Nlrp3 inflammasome effects and disregarding other effects of UAF1 complexes. For instance, both Fig 1a and Fig 1d show statistically significant effects of Uaf1 deficiency on IL-6 and TNF levels that are not mentioned in the results. As another example, lines 107 and 121 conclude only about effects on Nlrp3 expression, while the data show effects on the expression of multiple inflammatory genes. In addition, the new Fig 7i shows that the in vivo effects of Uaf1 deficiency reside at least partially at the level of pro-IL-1b expression, but this is not acknowledged in the text. I would like to ask the authors once more to please describe the wide-reaching effects of the UAF1 complexes in a balanced manner.

Related to this, the novel experiment I requested to evaluate the effect of UAF1 on Nlrc4- and Aim2-induced IL-1b secretion showed (as expected) that in addition to enhancing Nlrp3 inflammasome induced IL-1b secretion, UAF1 also potentiated the cytokine output of Nlrc4 and Aim2 inflammasome activation. Also this UAF1 effects should be acknowledged more clearly in the manuscript. The authors inserted this novel experiment as Fig S5c, but I think it would fit better in Fig S4 together with the current Fig 2c. This way, there would be one comprehensive figure showing that Uaf1 complexes do not influence Nlrc4 and Aim2 expression levels but do influence Nlrc4- and Aim2-induced IL1b production.

Finally, there is a mistake on lines 118-119. The authors state that 'Moreover, Uaf1 deficiency enhanced LPS-induced Il1b mRNA expression (Supplementary Fig. 3b)', while this figure shows the exact opposite: Uaf1-deficient cells produced less IL1b upon LPS treatment.

Reviewer #2:

In my previous recommendations to the authors, I had asked to 'be a bit less biased and re-write the manuscript (especially the abstract and discussion) to acknowledge the differential contributions of NF- κ B activation effects and Nlrp3 protein stability effects in regulating Nlrp3 inflammasome outcomes'. Instead of re-writing 'especially' the abstract and the discussion, the authors decided to re-write 'only' the abstract and the discussion (by adding 1 sentence to each). The manuscript continues to mislead the reader by focusing on Nlrp3 inflammasome effects and disregarding other effects of UAF1 complexes. For instance, both Fig 1a and Fig 1d show statistically significant effects of Uaf1 deficiency on IL-6 and TNF levels that are not mentioned in the results. As another example, lines 107 and 121 conclude only about effects on Nlrp3 expression, while the data show effects on the expression of multiple inflammatory genes. In addition, the new Fig 7i shows that the in vivo effects of Uaf1 deficiency reside at least partially at the level of pro-IL-1b expression, but this is not acknowledged in the text. I would like to ask the authors once more to please describe the wide-reaching effects of the UAF1 complexes in a balanced manner.

Answer: We are very sorry for misunderstanding your meaning. Indeed, the expression of TNF- α , IL-6 and pro-IL-1 β induced by NF- κ B activation is as important as NLRP3, which we ignored previously. We appreciated for the valuable advice and re-wrote abstract, results and discussion.

Related to this, the novel experiment I requested to evaluate the effect of UAF1 on Nlrc4- and Aim2-induced IL-1b secretion showed (as expected) that in addition to enhancing Nlrp3 inflammasome induced IL-1b secretion, UAF1 also potentiated the cytokine output of Nlrc4 and Aim2 inflammasome activation. Also this UAF1 effects should be acknowledged more clearly in the manuscript. The authors inserted this novel experiment as Fig S5c, but I think it would fit better in Fig S4 together with the current Fig 2c. This way, there would be one comprehensive figure showing that Uaf1 complexes do not influence Nlrc4 and Aim2 expression levels but do influence Nlrc4- and Aim2-induced IL1b production.

Answer: We accepted the valuable suggestion and rearranged the structure of manuscript. To make the description of UAF1 effects more completely and clearly, we put the expression level of NLRC4 and AIM2 together with the secretion of IL-1 β induced by NLRC4 and AIM2. In this situation, the effects of UAF1 on NF- κ B activation could be demonstrated more convincingly. To match this structure of manuscript, we inserted the new data in Fig S3a, together with the current Fig 2c.

Finally, there is a mistake on lines 118-119. The authors state that 'Moreover, Uaf1 deficiency enhanced LPS-induced Il1b mRNA expression (Supplementary Fig. 3b)', while this figure shows the exact opposite: Uaf1-deficient cells produced less IL1b upon LPS treatment.

Answer: We are very sorry for this mistake and have revised this statement in the new version.

REVIEWERS' COMMENTS

Reviewer #2 (Remarks to the Author):

I thank the authors for taking my comments into consideration. This manuscript can now be accepted.

Reviewer #2:

I thank the authors for taking my comments into consideration. This manuscript can now be accepted.

Answer: Thank you very much for all of the valuable advice. We very appreciate your patient guidance.